# ProGO: Probabilistic Global Optimizer

## Abstract

In the field of global optimization, many existing algorithms face challenges posed by non-convex target functions and high computational complexity or unavailability of gradient information. These limitations, exacerbated by sensitivity to initial conditions, often lead to suboptimal solutions or failed convergence. This is true even for Metaheuristic algorithms designed to amalgamate different optimization techniques to improve their efficiency and robustness. To address these challenges, we develop a sequence of multidimensional integration-based methods that we show to converge to the global optima under some mild regularity conditions. Our probabilistic approach does not require the use of gradients and is underpinned by a mathematically rigorous convergence framework anchored in the nuanced properties of nascent optima distribution. In order to alleviate the problem of multidimensional integration, we develop a latent slice sampler that enjoys a geometric rate of convergence in generating samples from the nascent optima distribution, which is used to approximate the global optima. The proposed Probabilistic Global Optimizer (ProGO) provides a scalable unified framework to approximate the global optima of any continuous function defined on a domain of arbitrary dimension. Empirical illustrations of ProGO across a variety of popular non-convex test functions (having finite global optima) reveal that the proposed algorithm outperforms, by order of magnitude, many existing state-of-the-art methods, including gradient-based, zeroth-order gradient-free, and some Bayesian Optimization methods, in term regret value and speed of convergence. It is, however, to be noted that our approach may not be suitable for functions that are expensive to compute.

## 1 Introcution

Global optimization constitutes a critical research area within applied mathematics and numerical analysis, aiming to locate the global optima of target functions over a specified domain. This field has substantial applications across various sectors in machine learning, such as hyperparameter tuning (Snoek et al., 2012), signal processing (Liu et al., 2020), and black-box adversarial attacks (Ru et al., 2019). A global optimization problem (minimization) with unique minima can be formulated as

$$\boldsymbol{x}^* = \arg\min_{\boldsymbol{x} \in \Omega} f(\boldsymbol{x}), \tag{1}$$

where $f(\boldsymbol{x})$ is generally a continuous function defined over a domain $\Omega$ ($\subseteq \mathbb{R}^d$), that is a subset of the $d$-dimensional Euclidean space $\mathbb{R}^d$. Extensive progress has been made in optimizing globally convex functions over compact domains, where the global minima $\boldsymbol{x}^*$ is guaranteed to be identified. However, less generalizable solutions exist for non-convex functions or on non-compact sets, even when some target function possesses smoothness or differentiability.

For semantic precision, we differentiate between "optimum" / "minimum", the optimal / lowest function value $f^*$, and "optima" / "minima", which corresponds to the $\boldsymbol{x}^*$ at which $f^*$ is attained. In this paper, we assume the existence of a finite $f^* = \min_{\boldsymbol{x} \in \Omega} f(\boldsymbol{x})$ and a non-empty set of minima $\Omega^* = \{\boldsymbol{x} \in \Omega : f(\boldsymbol{x}) = f^*\}$. When the minima $\boldsymbol{x}^*$ in eq. (1) is unique, $\Omega^*$ will be a singleton set. Importantly, $\Omega^*$ can comprise either a finite or infinite number of elements. One of our main contributions lies in the identification of the limitations of gradient-based techniques, particularly for non-convex functions, and the introduction of a reliable, integration-based alternative that guarantees to locate the global optima without convex assumptions.

Additionally, the efficacy of many global optimization algorithms is sensitive to initial points. Even metaheuristic algorithms, which amalgamate various optimization techniques for robustness, can yield suboptimal outcomes with poorly chosen initial conditions. Our method, under mild regularity conditions, is robust to initial conditions and yields accurate estimates of $x^*$ within a decent computational timeframe, provided that function evaluations are not expensive.

**Gradient-based algorithms.** Gradient-based methods like stochastic gradient descent (Robbins and Monro, 1951), Adam (Kingma and Ba, 2015), AdaGrad (Duchi et al., 2011), and RMSprop (Tieleman and Hinton, 2012) have found broad applications across disciplines. They have demonstrated their utility in various successful applications such as generative adversarial networks (Seward et al., 2018) and reinforcement learning (Mnih et al., 2016). While these methods offer practical effective solutions, their theoretical convergence to global optima is often framed within specific contexts, particularly when the target function $f(\boldsymbol{x})$ is smooth and convex. Recent variants like AMSGrad (Reddi et al., 2019) and parameter-selective approaches (Shi et al., 2020) explore parameter effects on theoretical guarantee and practical efficacy.

**Zeroth order (ZO) methods.** In cases where gradient information is unavailable, noisy, or computationally expensive to evaluate, such as signal processing and machine learning (Liu et al., 2020), ZO methods have emerged driven by the need to solve these problems. These techniques, also known as "black-box" or "derivative-free" optimization, bypass the need for gradients and focus solely on function values at any given point (Larson et al., 2019; Rios and Sahinidis, 2013). Several noteworthy methods have been proposed in this vein, including Gradientless Descent (GLD) (Golovin et al., 2019) which is numerically stable via a geometric approach, Random Gradient-Free (RGF) method via finite difference along a random direction by Nesterov and Spokoiny (2017), and Prior-Guided Random Gradient-Free (PRGF) by Cheng et al. (2021), typically operating under a framework that assumes convexity in the target functions. Additionally, Shu et al. (2022) introduced the Zeroth-Order Optimization with the Trajectory-Informed Derivative Estimation (ZoRD) algorithm, further enriching the query-efficient ZO optimization methods landscape.

**Global optimization challenges.** While gradient-based methods and ZO methods offer advantages, each comes with its set of limitations and is often contingent upon wise selections of initial parameters and starting points. Besides, Monte Carlo-based methods provide consistent global convergence (Harrison, 2010) but can be computationally demanding in high-dimensional spaces. Bayesian optimization techniques such as the Gaussian Process-Upper Confidence Bound (GP-UCB) algorithm (Srinivas et al., 2009) and Trust Region Bayesian Optimization (TuRBO) (Eriksson et al., 2019) assume $f(\boldsymbol{x})$ follows the Gaussian Process, whose performances hinge upon careful selection of acquisition and kernel functions. Analytical methods do contribute to domain-specific solutions but can entail intricate numerical challenges, affecting their widespread applicability (Corriou and Corriou, 2021). Most literature in this field has been oriented towards establishing first-order optimality conditions, often under function convexity and differentiability assumptions. A notable work by Luo (2018) formalized a rigorous mathematical relation between an arbitrary continuous function $f$ defined over a compact set $\Omega \subseteq \mathbb{R}^d$ and its corresponding global minima $f^*$; however, this work only built a theoretical framework. This underscores the pressing need for an efficient and robust global optimization framework, especially in addressing non-convex and high-dimensional challenges. Our approach stands out significantly different from model-based methodologies ((Hu et al., 2012; Zhang et al., 2018)). Model-based methods typically provide approximations of optimality through a two-step process: starting with an initial modeling assumption that approximates the target function and subsequently deriving estimators based on the assumed model. Our proposed method requires no such sugrrorgate function to approximate the target function but rather is based on the given target function.

**Main contribution.** This paper introduces the Probabilistic Global Optimizer (ProGO), a novel non-gradient-based global optimization algorithm based on a sequence of sampling from a suitable probability distribution. Our work significantly extends the theoretical framework laid by Luo (2018), notably in three key dimensions:

1. **Generalization to Non-Compact Set:** Luo (2018)'s work is based on the assumption that $\Omega$ is a compact set. Such an assumption may limit the scope of its applicability to a class

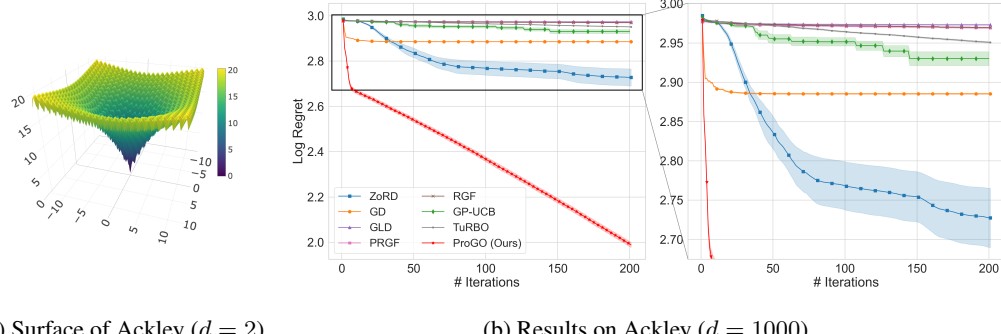

(a) Surface of Ackley ($d = 2$)          (b) Results on Ackley ($d = 1000$)

Figure 1: (a) Visualization of the Ackley function for $d = 2$. (b) Evaluation of ProGO and competing methods (elaborated upon in Section 4) applied to the Ackley function in a high-dimensional setting with $d = 1000$. The $x$-axis represents the iteration count, while the $y$-axis denotes the average log-scaled regret. Each curve shows the mean $\pm$ standard error across ten independent runs. ProGO outperforms other methods by a notably faster rate of convergence accompanied by smaller variability.

    of popular functions when $\Omega = \mathbb{R}^d$. E.g., even when $d = 1$, the elementary function $f(\boldsymbol{x}) = \boldsymbol{x}^2$ defined over $\mathbb{R}$ is unbounded. We generalize the domain to an arbitrary subset of $\mathbb{R}^d$ by incorporating a sequence of probability distribution on $\mathbb{R}^d$ that assigns less and less mass outside $\Omega^*$ and eventually converging with support $\Omega^*$.

2. **A Practically Efficient Algorithm:** While a theoretical framework is established in Luo (2018) with several interesting results when the domain $\Omega$ is compact and the function $f$ is assumed continuous or smooth (with second order derivatives), to the best of our knowledge, no practical methods are yet available to evaluate the (potentially high-dimensional) integration required to estimate the minima. We fill this gap by developing a practically efficient algorithm that we call ProGO, which uses a latent slice sampler (explained later) to efficiently obtain samples from the probability distribution of the minima.

3. **Extensive Experiments Validation:** We carried out a comprehensive series of experiments to evaluate ProGO's performance in comparison to various types of leading global optimization techniques, including Gradient Descent (GD), Zeroth-Order Optimization (ZO), and Bayesian Optimization (BO). Our empirical evidence demonstrates that ProGO consistently surpasses all the algorithms we compared against across various metrics – most notably, geometric rate of convergence to global minima and computational efficiency as indicated by function evaluations or CPU time. Specifically, we illustrate the superior numerical performance of our proposed ProGO for the popular Ackley function (known to have several local optima with a global minimum at the origin) for dimensions ranging from $d = 20$ to $d = 1000$ (refer to Fig. 1), as well as the Levy function (refer to Fig. 3). As depicted in the figures, the logarithmic regret exhibits a linear rate of convergence, which in the original scale translates to geometric convergence, outpacing the majority of extant global optimization algorithms.

The rest of this paper unfolds as follows: Section 2 lays the theoretical groundwork on the probabilistic minima distribution for our approach; Section 3 details the latent slice sampler and our ProGO algorithm; Section 4 presents the empirical validations, and Section 5 concludes the paper.

## 2 PRELIMINARIES

### 2.1 NASCENT MINIMA PROBABILITY DISTRIBUTION

Our proposed algorithm, ProGO, is based on generalizing the sequence of nascent minima (probability) distributions defined in Luo (2018) using an arbitrary (prior) probability measure with full support on the Euclidean space $\mathbb{R}^d$. This distribution possesses advantageous properties that will be elaborated upon in subsequent discussions. In particular, we will show how to efficiently generate

samples from the sequence of such distributions and subsequently use the empirical (posterior) mean and other summaries to estimate the minimum value $f^*$ and the minima $\boldsymbol{x}^*$.

**Assumption 1.** *The following conditions are assumed throughout the paper:*

(i) *Assume that the function $f : \Omega \subseteq \mathbb{R}^d \to \mathbb{R}$ is a continuous function with a finite global minimum value $f^*$; i.e., $f(\boldsymbol{x}) \geqslant f^*$ for all $\boldsymbol{x} \in \Omega$.*

(ii) *The set of global minima $\Omega^* = \{\boldsymbol{x} \in \Omega : f(\boldsymbol{x}) = f^*\}$ is non-empty.*

(iii) *There is a probability measure with density $\pi(\boldsymbol{x})$ that has full suppprt on $\mathbb{R}^d$. In other words, $\pi(\boldsymbol{x}) > 0$ for any $\boldsymbol{x} \in \mathbb{R}^d$ and $\int_{\mathbb{R}^d} \pi(\boldsymbol{x}) d\boldsymbol{x} = 1$. Here, the integration is with respect to the Lebesgue measure on $\mathbb{R}^d$.*

In the above, the probability density $\pi(\boldsymbol{x})$ can be chosen arbitrarily, but in practice, we can use a uniform distribution when $\Omega$ is compact, or a very flat (nearly uniform) distribution when $\Omega$ is unbounded. Regardless, we next define a nascent minima distribution when it depends on the choice of the density $\pi(\cdot)$.

**Definition 1.** *Nascent Minima (probability) distribution:*

*For any $k \geqslant 0$, a nascent minima distribution density is defined as:*

$$m_k(\boldsymbol{x}) = \frac{e^{-kf(\boldsymbol{x})} \cdot \pi(\boldsymbol{x})}{\int_\Omega e^{-kf(\boldsymbol{t})} \cdot \pi(\boldsymbol{t})\,\mathrm{d}\boldsymbol{t}}. \tag{2}$$

**Remark 1.** *Note that the denominator in eq. (2) is a finite positive quantity for any arbitrary $k \geqslant 0$, because $0 < \int_\Omega e^{-kf(\boldsymbol{t})} \cdot \pi(t)\,\mathrm{d}t \leqslant e^{-kf^*}$. The assumption that $\pi(\cdot)$ is a probability density can be relaxed as long as $\int_\Omega e^{-kf(\boldsymbol{t})} \cdot \pi(\boldsymbol{t})\,\mathrm{d}\boldsymbol{t} < \infty$ for any $k > 0$, even when $\int_\Omega \pi(t)\,\mathrm{d}t = \infty$.*

Notice that $m_k(\boldsymbol{x})$ can also written as

$$m_k(\boldsymbol{x}) = \frac{e^{-k(f(\boldsymbol{x})-f^*)} \cdot \pi(\boldsymbol{x})}{\int_\Omega e^{-k(f(\boldsymbol{t})-f^*)} \cdot \pi(\boldsymbol{t})\,\mathrm{d}\boldsymbol{t}}. \tag{3}$$

By replacing the original $f(\boldsymbol{x})$ by $f(\boldsymbol{x}) - f^*$, we may assume without loss of generality that $f$ is a non-negative valued function and has a global minimum $f^* = 0$. Consequently, we focus on finding a solution in the set of global minima $\Omega^* = \{\boldsymbol{x} \in \Omega : f(\boldsymbol{x}) = 0\} \neq \emptyset$ for the rest of this paper for all subsequent theoretical analyses.

However, it should be noted that for practical applications, we need to work with the original function $f$ as we will not know that global minimum $f^*$ and set of minima $\Omega^*$, and our goal would be to approximate $f^*$ and $\boldsymbol{x}^* \in \Omega^*$ by letting $k \to \infty$.

**Remark 2.** *If the original $f$ is a positive valued function with $f^* > 0$, we can replace it with $\log f$ when defining the nascent minima density in eq. (2). Also, if a global maximum is desired, we can replace the original $f$ by $-f$ in defining the nascent minima density in eq. (2).*

Next, we provide two results under very minimal conditions, which establish the convergence of the (generalized) moments of the nascent minima distribution to the minimum value $f^*$. With additional conditions, we also establish the convergence of the minima.

## 2.2 CONVERGENCE PROPERTIES

In this subsection, we establish the theoretical underpinnings regarding the convergence properties of the nascent minima distribution.

**Theorem 1.** *Consider a function $f$ and a probability density $\pi$ satisfying the assumptions (i)-(iii) given in Assumption 1. Then, the nascent minima distribution has the following properties:*

$$\lim_{k \to \infty} \int_\Omega f(\boldsymbol{x}) m_k(\boldsymbol{x})\mathrm{d}\boldsymbol{x} = f^* \tag{4}$$

The proof is detailed in Section A.1. The above result implies that the expected value $\mathbb{E}_{m_k}[f(X)]$ converges to the minimum value $f^*$ and hence, if we are able to generate samples from the nascent

minima distribution $m_k(\cdot)$ for any $k > 0$, then we can approximate $f^*$ arbitrarily close by choosing a large $k > 0$. Next, we show that the above convergence is monotonic, which in turn implies that by increasing $k$ sequentially, we will get closer and closer to the minimum value $f^*$.

**Theorem 2** (monotonicity). *Consider a non-constant function $f$ and a probability density $\pi$ satisfying the assumptions (i)-(iii) given in Assumption 1. For each $k > 0$, let $\mu_k = \mathbb{E}_{m_k}[f(X)]$ denotes the expectation of $f(X)$ when $X \sim m_k(\cdot)$. Then the sequence $\{\mu_k\}$ is monotonically decreasing and satisfies*

$$\frac{\mathrm{d}\mu_k}{\mathrm{d}k} = \frac{\mathrm{d}}{\mathrm{d}k} \int_\Omega f(\boldsymbol{x}) m_k(\boldsymbol{x}) \mathrm{d}\boldsymbol{x} = -\mathbb{V}ar_k(f) < 0, \tag{5}$$

*where $\mathbb{V}ar_k(f) = \int_\Omega \left(f(\boldsymbol{x}) - \mu_k\right)^2 m_k(\boldsymbol{x})\mathrm{d}\boldsymbol{x}$ denotes the variance of $f(X)$ when $X \sim m_k(\cdot)$.*

The detailed proof is provided in Section A.2. Notice that the monotonic convergence of $\mu_k$ as $k$ increases is established for any continuous function $f$ and probability density $\pi$ satisfying (i)-(iii) of Assumption 1. This allows us for a very general use of ProGO with minimal assumptions for any dimension $d \geqslant 1$. Next, we explore the convergence of the minimum values with additional assumptions.

Here, we introduce the strong separability condition to define the scenario in which this ProGo method is most suitable.

**Assumption 2** (strong separability condition). *Consider the set of minima $\Omega^* = \{\boldsymbol{x} \in \Omega : f(\boldsymbol{x}) = f^*\}$ which is assumed to be non empty. Then $f$ is said to satisfy a strong separability if, for any given $\delta > 0$, we have $\inf_{\boldsymbol{x} \notin \Omega^* : ||\boldsymbol{x} - \tilde{\boldsymbol{x}}|| > \delta} f(\boldsymbol{x}) > f^*$ for any $\tilde{\boldsymbol{x}} \in \Omega^*$.*

The above condition implies that if $\boldsymbol{x} \notin \Omega^*$ and $||\boldsymbol{x} - \tilde{\boldsymbol{x}}|| > \delta$ for some $\delta > 0$, then $f(\boldsymbol{x}) > f^*$. In other words, the $f$ values for $\boldsymbol{x}$ not in $\Omega^*$ are well separated from those that are in $\Omega^*$ and hence $\epsilon_0 = \inf\{f(\boldsymbol{x}) - f^* : \boldsymbol{x} \notin \Omega^*\} > 0$.

**Theorem 3.** *Consider a bounded probability density $\pi$ and a target function satisfying the assumptions (i)-(iii) in Assumption 1. For each $k > 0$, let $\Omega_k = \{\boldsymbol{x} \in \Omega : m_k(\boldsymbol{x}) \geqslant m_k(\tilde{\boldsymbol{x}}), \forall \tilde{\boldsymbol{x}} \in \Omega\}$ denotes the set of maximizes for $m_k(\boldsymbol{x})$. Then for any sequence $\{\boldsymbol{x}_k^*\}$, $\boldsymbol{x}_k^* \in \Omega_k$, it satisfy*

$$\lim_{k \to \infty} f(\boldsymbol{x}_k^*) = f^*, \tag{6}$$

*In addition, if the target function satisfies the strong separability condition in Assumption 2, we have*

$$\lim_{k \to \infty} \inf_{\tilde{\boldsymbol{x}} \in \Omega^*} ||\boldsymbol{x}_k^* - \tilde{\boldsymbol{x}}|| = 0. \tag{7}$$

The proof is available in Section A.3. This theorem establishes the convergence of the sequence $\boldsymbol{x}_k^*$ toward the global minima set $\Omega^*$ by quantifying the metric $\inf_{\tilde{\boldsymbol{x}} \in \Omega^*} ||\boldsymbol{x}_k^* - \tilde{\boldsymbol{x}}||$. This metric serves as a measure of divergence between the iteratively obtained minima $\boldsymbol{x}_k^*$ for each $k$ and elements from the true global minima set $\Omega^*$. The empirical evaluations in Section 4 corroborate the algorithm's efficacy in identifying discrepancies in both the optimal value and corresponding minima.

## 3 ProGO: A New Probabilistic Global Optimization Method

The rationale for the sampling procedure of ProGO is based on the fact that the mode (and also the mean) of the density $m_k(x)$ converges to the minima when it is unique. Consequently, by sampling random variates from this density, we estimate the mode and the mean using Monte Carlo methods, as exact analytical calculations are often impractical. We dicuss the Latent Slice Sampler (LSS) that we adoped with a one dimensional illustration in Section 3.1, followed by the description of the ProGO algorithm in Section 3.2.

### 3.1 Latent Slice Sampler

Compared to traditional Markov Chain Monte Carlo techniques like the Metropolis–Hastings algorithm, slice sampling (SS) provides benefits such as reduced asymptotic variance and accelerated

convergence (Mira and Tierney, 2002; Neal, 2003; Roberts and Rosenthal, 1999). From several SS variants such as Elliptical SS by Murray et al. (2010) and Polar SS by Rudolf and Schär (2023), we adopt the LSS introduced by Li and Walker (2023) because it eliminates the requirement for proposal distribution and improves efficiency in high-dimensional sampling.

For a detailed understanding of LSS, consider the target distribution as a minima distribution $m(\boldsymbol{x})$ for a $d$-dimensional variable $\boldsymbol{x}$. By incorporating slice variables $w$, $\boldsymbol{s} = (s_1, \cdots, s_d)^\top$, and $\boldsymbol{l} = (l_1, \cdots, l_d)^\top$, the joint density can be formulated as follow:

$$p(\boldsymbol{x}, w, \boldsymbol{s}, \boldsymbol{l}) = \mathbb{I}\left(m(\boldsymbol{x}) > w\right) p(\boldsymbol{s}) \prod_{j=1}^{d} \frac{\mathbb{I}\left(\boldsymbol{x}_j - \boldsymbol{s}_j/2 < \boldsymbol{l}_j < \boldsymbol{x}_j + \boldsymbol{s}_j/2\right)}{\boldsymbol{s}_j}. \tag{8}$$

Each $\boldsymbol{s}_j$ for $j = 1, \cdots, d$ is assumed to follow an independent gamma distribution with a shape parameter of 2 and scale parameter $\beta$ of 5, following Luo (2018). Let $\boldsymbol{x}^{(t)}$ represent the sample obtained after $t^{th}$ iteration, the full LSS algorithm implemented via Gibbs sampling is presented in Algorithm 2.

**One Dimensional Illustration.** To demonstrate the rationale behind using the nascent minima distribution for global optimization and to evaluate the efficacy of LSS, we consider a one-dimensional example adapted from Luo (2018), given by $f(x) = \cos(x^2) + x/5 + 1, x \in [0, 5]$. This function possesses three local minima and a singular global minima at $f(x) = 0.353$ when $x = 1.756$.

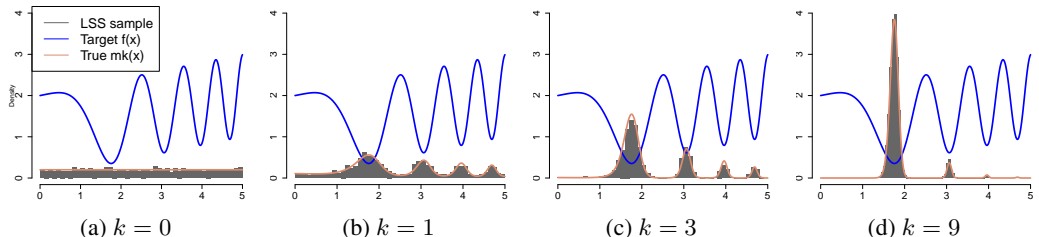

| (a) $k = 0$ | (b) $k = 1$ | (c) $k = 3$ | (d) $k = 9$ |

Figure 2: One dimensional illustration of nascent minima distribution $m_k(\boldsymbol{x})$ and its latent slice samples. In this illustration, $\pi(\boldsymbol{x})$ is assumed to satisfy Assumption 1 and uniformly distributed within [0,5]. The target function $f(x) = \cos(x^2) + x/5 + 1$ is depicted in the blue curve. The true $m_k(\boldsymbol{x})$ with $k = 0, 1, 3, 9$ are shown as a red curve. The black histograms represent the latent slice samples of $m_k(\boldsymbol{x})$.

As depicted in Figure 2, the density distributions generated through LSS closely align with the actual minima distribution across various $k$. Figure 2 also visualizes the impact of increasing $k$ from 0 to 1, 3, and 9. As $k$ increases, the density of mk(x) converges towards the global optima of f(x), resulting in more accurate sampling around the global optima.

Notably, when $k = 0$, $m_0(\boldsymbol{x}) = \pi(\boldsymbol{x})$ corresponds to a uniform distribution, as shown in Figure 2(a). A higher value of $k$ leads to a minima distribution where the probability is increasingly focused on the global optima.

### 3.2 PROGO

We outline the algorithm of ProGO in Algorithm 1. The step size is incremented by $\triangle k$ for each iteration, chosen as $\triangle k = (e - 1)k$. Such choice is motivated by the inverse shrinkage rate of $m_k(\boldsymbol{x})$ with respect to $k$, as demonstrated in Luo (2018). The starting value of $k$ establishes the subsequent sequence $\{k, ke, ke^2, \cdots, ke^T\}$ across $T$ iterations, leading to a converging sequence of $\{f(\boldsymbol{x}_k^*), f(\boldsymbol{x}_{ke}^*), f(\boldsymbol{x}_{ke^2}^*), \cdots, f(\boldsymbol{x}_{ke^T}^*)\}$, where $x_k^* \in \{\boldsymbol{x} \in \Omega : m_k(\boldsymbol{x}) \geqslant m_k(\tilde{\boldsymbol{x}}), \forall \tilde{\boldsymbol{x}} \in \Omega\}$. Based on our preliminary results, the initial value of $k$ is set to 5.

Notice that ProGO does not necessitate prior knowledge of $f^*$ or any assumptions about the target function $f$ other than continuity and the existence of the minimum value. Besides, the output of ProGO delivers more than just an optimum value; it also provides the sample sets from the minima

---

**Algorithm 1:** ProGO Algorithm

---

**Input:** Target function $f : \Omega \to \mathbb{R}$, probability distribution function $\pi(\cdot)$, max iteration number $T = 200$, sample size $N$, starting point $\boldsymbol{x}^{(0)}$, burn-in period $n_b$

1 Initialize $k = 5$
2 **for** iteration $t \in \{1, 2, \cdots, T\}$ **do**
3 $\quad$ $\boldsymbol{x}^{(1)}, \cdots, \boldsymbol{x}^{(N)} \sim \text{LSS-ProGO}(m_k(\cdot\,; \pi, f), N, \boldsymbol{x}^{(0)}, n_b)$
4 $\quad$ $\tilde{\boldsymbol{x}}^{(t)} \leftarrow \arg\max_{i \in \{1, 2, \cdots, N\}} m_k(\boldsymbol{x}^{(i)}; \pi, f))$
5 $\quad$ $k \leftarrow k + (e-1)k$
6 **end**

**Output:** $\arg\min_{t \in \{1, 2, \cdots, T\}} f(\tilde{\boldsymbol{x}}^{(t)})$

---

distribution. This provides valuable information about the distributional properties of local minima, as exemplified in Figure 2.

## 4 EXPERIMENTS

Test functions, evaluation metrics, and benchmarking methods play crucial roles in validating the performance of an optimization algorithm. While a wide array of test functions exists in the literature (Jamil and Yang, 2013), the Ackley function—formulated initially by Ackley (2012)—and the Levy function remain prevalent choices, as corroborated by recent studies like Shu et al. (2022).

As for evaluation criteria, we use the following metrics of function log regret and minima log regret to capture discrepancies in both $f(\boldsymbol{x})$ and $\boldsymbol{x}$.

**Definition 2.** *Given $\tilde{\boldsymbol{x}}$ as an estimated optima in a $d$-dimension space, the **function log regret** is defined as:*

$$r_f = \log\left(f(\tilde{\boldsymbol{x}}) - f^*\right), \tag{9}$$

*quantifying the deviation between the estimated and true global optimum. The **minima log regret** is formulated as:*

$$r_m = \log \frac{||\tilde{\boldsymbol{x}} - \boldsymbol{x}^*||}{\sqrt{d}}, \tag{10}$$

*which quantifies the discrepancy between the estimated and true global minima.*

The experimental design and the selection of competing algorithms are mostly aligned with the framework presented in ZoRD (Shu et al., 2022) to ensure a consistent and fair evaluation (details in Section D). A computational budget capped at 200 iterations is allotted for each of the ten independent runs conducted in every experiment setting. We chose the value 200 because we found in preliminary results that our method's regrets had already reached an extremely small value of $1 \times 10^{-17}$, far less than the threshold that can be accurately represented by floating-point numbers. The evaluated methods include: **1) ZoRD**: Zeroth-order trajectory-informed derivative estimation (Shu et al., 2022). **2) GD**: Gradient-Descent, directly using first-order information. **3) GLD**: Gradientless Descent (Golovin et al., 2019). **4) PRGF**: Prior-guided random gradient-free algorithm (Cheng et al., 2021). **5) RGF**: Random gradient-free method via finite difference (Nesterov and Spokoiny, 2017). **6) GP-UCB**: Gaussian Process-Upper Confidence Bound (Srinivas et al., 2009). **7) TuRBO**: Trust Region Bayesian Optimization (Eriksson et al., 2019). **8) ProGO (our approach)**: Probabilistic Global Optimization.

### 4.1 ACKLEY

The Ackley function serves as a prominent benchmark for evaluating optimization algorithms. It is characterized as a continuous, differentiable, multimodal, and non-convex function, thereby posing significant optimization challenges. Mathematically, the $d$-dimensional Ackley function is given by:

$$f(\boldsymbol{x}) = -a\exp\left(-b\sqrt{\frac{1}{d}\sum_{i=1}^{d}\boldsymbol{x}_i^2}\right) - \exp\left(\frac{1}{d}\sum_{i=1}^{d}\cos\left(c\boldsymbol{x}_i\right)\right) + a + \exp(1), \tag{11}$$

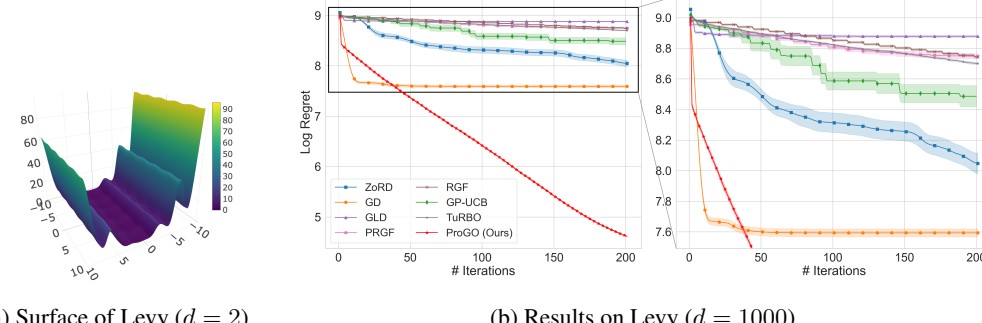

(a) Surface of Levy ($d = 2$)  (b) Results on Levy ($d = 1000$)

Figure 3: (a) Visualization of the Levy function for $d = 2$. (b) Evaluation of ProGO and competing methods applied to the Levy function in a high-dimensional setting with $d = 1000$. The $x$-axis represents the iteration count, while the $y$-axis denotes the average log-scaled regret. Each curve shows the mean $\pm$ standard error across ten independent runs.

where recommended parameter values are $a = 20, b = 0.2$ and $c = 2\pi$ (Adorio and Diliman, 2005). As shown in Figure 1 (a), the Ackley function features numerous local minima and a central global minima at $f^* = 0$ and $\boldsymbol{x}^* = (0, \cdots, 0)^\top$, presenting multiple local minima traps for optimization algorithms.

Table 1: Comparison of performance on the Ackley function for dimensions $d = 20, 40, 1000$: Each optimization method has ten independent runs. Accuracy is quantified using the average function log regret ($r_f$) and the average minima log regret ($r_m$) across these ten runs. Computational efficiency is represented by the average runtime ($t$) across all ten runs, measured in seconds.

| | $d = 20$ | | | $d = 40$ | | | $d = 1000$ | | |
|---|---|---|---|---|---|---|---|---|---|
| Method | $r_f$ | $r_m$ | $t$ (s) | $r_f$ | $r_m$ | $t$ (s) | $r_f$ | $r_m$ | $t$ (s) |
| ZoRD | 2.83 | 1.68 | 1265.43 | 2.85 | 1.72 | 1092.04 | 2.73 | 1.58 | 406.16 |
| GD | 2.85 | 1.72 | 0.12 | 2.87 | 1.75 | 0.13 | 2.89 | 1.75 | 0.14 |
| GLD | 2.73 | 1.65 | 0.38 | 2.89 | 1.70 | 0.39 | 2.97 | 1.75 | 0.40 |
| PRGF | 2.85 | 1.70 | 0.06 | 2.89 | 1.74 | 0.07 | 2.97 | 1.75 | 0.08 |
| RGF | 2.86 | 1.66 | 0.06 | 2.89 | 1.72 | 0.07 | 2.97 | 1.74 | 0.08 |
| GP-UCB | 2.01 | 1.62 | 132.13 | 2.07 | 1.63 | 323.12 | 2.93 | 1.74 | 1132.61 |
| TuRBO | 1.73 | 1.60 | 30.83 | 2.54 | 1.62 | 83.84 | 2.95 | 1.73 | 272.56 |
| **ProGO** | **-35.35** | **-35.86** | 6.77 | **-31.56** | **-9.50** | 16.25 | **1.99** | **0.52** | 280.05 |

Our empirical evaluations (see Table 1) span dimensions $d = 20, 40$, and $1000$. Across all dimensions, ProGO consistently outperforms other methods, achieving significantly lower function log regret and lower minima log regret. Moreover, Figure 1(b) corroborates ProGO's geometric rate of convergence even on high dimensions (see Figure 5 for results on $d = 20$ and $d = 40$).

## 4.2 LEVY

The Levy function is another frequently used test function in optimization research, as in the work of Shu et al. (2022). It is a continuous and non-convex function defined as:

$$f(\boldsymbol{x}) = \sin^2(\pi w_1) + \sum_{i=1}^{d-1} (w_i - 1)^2 \left[1 + 10\sin^2(\pi w_i + 1)\right] + (w_d - 1)^2 \left[1 + \sin^2(2\pi w_d)\right], \quad (12)$$

where $w_i = 1 + (\boldsymbol{x}_i - 1)/4, \forall i = 1, \ldots, d$. The global minimum is $f^* = 0$ attained at $\boldsymbol{x}^* = (1, \ldots, 1)^\top$. As shown in Figure 3, the Levy function presents a more complex optimization challenge than the Ackley function due to the substantially flatter area that surrounds its global optima.

Table 2: Comparison of performance on the Levy function for dimensions $d = 40, 100, 1000$: Each optimization method undergoes ten independent runs. Accuracy is quantified using the average function log regret ($r_f$) and the average minima log regret ($r_m$) across these ten runs. Computational efficiency is represented by the average runtime ($t$) across all ten runs, measured in seconds.

| Method | $d = 40$ | | | $d = 100$ | | | $d = 1000$ | | |
|--------|-------|-------|---------|-------|-------|---------|-------|-------|---------|
| | $r_f$ | $r_m$ | $t$ (s) | $r_f$ | $r_m$ | $t$ (s) | $r_f$ | $r_m$ | $t$ (s) |
| ZoRD | 4.08 | 1.56 | 585.69 | 5.26 | 1.56 | 1514.97 | 8.05 | 1.58 | 272.29 |
| GD | 4.10 | 1.58 | 0.16 | 5.15 | 1.61 | 0.15 | 7.59 | 1.59 | 0.14 |
| GLD | 4.89 | 1.59 | 0.40 | 6.15 | 1.60 | 0.39 | 8.88 | 1.59 | 0.39 |
| PRGF | 4.68 | 1.59 | 0.08 | 5.96 | 1.61 | 0.07 | 8.75 | 1.60 | 0.07 |
| RGF | 4.59 | 1.58 | 0.08 | 5.84 | 1.62 | 0.08 | 8.74 | 1.59 | 0.09 |
| GP-UCB | 3.80 | 1.58 | 216.83 | 5.28 | 1.58 | 651.38 | 8.49 | 1.58 | 1400.41 |
| TuRBO | 3.58 | 1.56 | 77.66 | 5.18 | 1.55 | 197.78 | 8.70 | 1.59 | 265.71 |
| **ProGO** | **-0.05** | **-0.96** | 11.85 | **1.55** | **-0.50** | 27.33 | **4.62** | **-0.01** | 311.99 |

Empirical evaluation of the Levy function for dimensions $d = 40, 100, 1000$ is presented in Table 2. Notably, ProGO demonstrates significantly lower log regrets relative to other competing methods across all dimensions. Furthermore, Figure 3 and Figure 6 show that ProGO exhibits a markedly faster convergence rate compared to competing gradient-based, zeroth-order, and Bayesian optimization methods. Notice that Gradient Descent, depicted in orange, initially exhibits rapid convergence but is then trapped in local optima.

## 5 CONCLUSION

In this paper, we introduce ProGO, a novel probabilistic global optimization algorithm leveraging minima distribution theory and latent slice sampling technique. Our methodology represents a significant departure from conventional gradient-based methods, offering robust convergence guarantees for global optima while preserving computational efficiency without using gradient information.

Specifically, our contributions are threefold: We extend Luo (2018)'s theoretical framework to noncompact sets and prove its global convergence. Based on the generalized framework, we implement the ProGO algorithm, integrating a latent slice sampler for enhanced computational efficiency, especially for high dimensions. Finally, comprehensive experiments demonstrate ProGO's outstanding performance over state-of-the-art methods in terms of accuracy and convergence speed on various functions and dimensions.

However, it is worth noting that ProGO may not be suitable for optimization problems where function evaluation is computationally expensive. Future investigations on enhancing the algorithm's computational efficiency and extending the applicability of ProGO to more diverse problem domains may contribute to the growing field of global optimization.

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

# A PROOFS

## A.1 PROOF FOR THEOREM 1

*Proof for Theorem 1.* Recall that $\Omega^* = \{\boldsymbol{x} \in \Omega : f(\boldsymbol{x}) = 0\}$ without loss of generalizability. For any $\epsilon > 0$, define $\Omega_\epsilon = \{\boldsymbol{x} \in \Omega : f(\boldsymbol{x}) < \epsilon\}$ and $\Omega_\epsilon^c = \Omega \setminus \Omega_\epsilon$, then

$$
\begin{aligned}
\int_\Omega f(\boldsymbol{x}) m_k(\boldsymbol{x}) d\boldsymbol{x} &= \frac{\int_{\Omega_{\frac{\epsilon}{2}}} f(\boldsymbol{x}) \cdot e^{-kf(\boldsymbol{x})} \pi(\boldsymbol{x}) \mathrm{d}\boldsymbol{x} + \int_{\Omega_{\frac{\epsilon}{2}}^c} f(\boldsymbol{x}) \cdot e^{-kf(\boldsymbol{x})} \cdot \pi(\boldsymbol{x}) \mathrm{d}\boldsymbol{x}}{\int_\Omega e^{-kf(t)} \cdot \pi(t) \mathrm{d}t} \\
&< \frac{\epsilon}{2} \cdot \frac{\int_{\Omega_{\frac{\epsilon}{2}}} e^{-kf(\boldsymbol{x})} \cdot \pi(\boldsymbol{x}) \mathrm{d}\boldsymbol{x}}{\int_\Omega e^{-kf(t)} \cdot \pi(t) \mathrm{d}t} + \frac{\int_{\Omega_{\frac{\epsilon}{2}}^c} f(\boldsymbol{x}) \cdot e^{-kf(\boldsymbol{x})} \cdot \pi(\boldsymbol{x}) \mathrm{d}\boldsymbol{x}}{\int_{\Omega_{\frac{\epsilon}{4}}} e^{-kf(t)} \cdot \pi(t) \mathrm{d}t} \\
&\leqslant \frac{\epsilon}{2} \cdot 1 + \frac{\int_{\Omega_{\frac{\epsilon}{2}}^c} f(\boldsymbol{x}) e^{-k\left(f(\boldsymbol{x}) - \frac{\epsilon}{4}\right)} \cdot \pi(\boldsymbol{x}) \mathrm{d}\boldsymbol{x}}{\int_{\Omega_{\frac{\epsilon}{4}}} \pi(t) \mathrm{d}t} \\
&= \frac{\epsilon}{2} + \int_{\Omega_{\frac{\epsilon}{2}}^c} \frac{f(\boldsymbol{x}) \pi(\boldsymbol{x}) e^{-k\left(f(\boldsymbol{x}) - \frac{\epsilon}{4}\right)}}{\int_{\Omega_{\frac{\epsilon}{4}}} \pi(t) \mathrm{d}t} \mathrm{d}\boldsymbol{x} \triangleq \frac{\epsilon}{2} + \int_{\Omega_{\frac{\epsilon}{2}}^c} g_k(\boldsymbol{x}) \mathrm{d}\boldsymbol{x}.
\end{aligned}
$$

The first inequality lies in $f(\boldsymbol{x}) < \frac{\epsilon}{2}$ for any $\boldsymbol{x} \in \Omega_{\frac{\epsilon}{2}}$ and $1/\int_\Omega e^{-kf(t)} \cdot \pi(t) \mathrm{d}t < 1/\int_{\Omega_{\frac{\epsilon}{2}}} e^{-kf(t)} \cdot \pi(t) \mathrm{d}t$. The second inequality is given by $\int_{\Omega_{\frac{\epsilon}{2}}} e^{-kf(\boldsymbol{x})} \cdot \pi(\boldsymbol{x}) \mathrm{d}\boldsymbol{x} < \int_\Omega e^{-kf(\boldsymbol{x})} \cdot \pi(\boldsymbol{x}) \mathrm{d}\boldsymbol{x}$ and $f(\boldsymbol{x}) < \frac{\epsilon}{4}$ for any $\boldsymbol{x} \in \Omega_{\frac{\epsilon}{4}}$. Define $g_k(\boldsymbol{x}) = \frac{f(\boldsymbol{x}) \pi(\boldsymbol{x}) e^{-k\left(f(\boldsymbol{x}) - \frac{\epsilon}{4}\right)}}{\int_{\Omega_{\frac{\epsilon}{4}}} \pi(t) \mathrm{d}t}$ for $\boldsymbol{x} \in \Omega_{\frac{\epsilon}{2}}^c$, then $\{g_k(\boldsymbol{x})\}_{k=1}^\infty$ is a sequence of nonnegative functions that monotonously decreases to 0 when $k$ goes to infinity and converges to zero, i.e., $\lim_{k\to\infty} g_k(\boldsymbol{x}) = 0$. Hence, by monotone

convergence theorem, $\lim_{k\to\infty}\int_{\Omega_{\frac{\epsilon}{2}}^c} g_k(\boldsymbol{x})\mathrm{d}\boldsymbol{x} = 0$. Consequently, for any $\epsilon > 0$, there exists a large $K_0$, such that $\int_{\Omega_{\frac{\epsilon}{2}}^c} g_k(\boldsymbol{x})\mathrm{d}\boldsymbol{x} < \epsilon/2$ for any $k > K_0$, and thus

$$0 \leqslant \int_\Omega f(\boldsymbol{x})m_k(\boldsymbol{x})d\boldsymbol{x} < \frac{\epsilon}{2} + \frac{\epsilon}{2} = \epsilon,$$

which proves $\lim_{k\to\infty}\int_\Omega f(\boldsymbol{x})m_k(\boldsymbol{x})\mathrm{d}\boldsymbol{x} = f^*(= 0)$.

$\square$

### A.2 PROOF FOR THEOREM 2

*Proof for Theorem 2.* For every $k \in \mathbb{R}$,

$$\begin{aligned}
\frac{\mathrm{d}}{\mathrm{d}k}\log m_k(\boldsymbol{x}) &= \frac{\mathrm{d}}{\mathrm{d}k}\left(-kf(\boldsymbol{x}) + \log\pi(\boldsymbol{x}) - \log\left\{\int_\Omega e^{-kf(\boldsymbol{t})}\cdot\pi(\boldsymbol{t})\,\mathrm{d}\boldsymbol{t}\right\}\right) \\
&= -f(\boldsymbol{x}) - \frac{\frac{\mathrm{d}}{\mathrm{d}k}\left\{\int_\Omega e^{-kf(\boldsymbol{t})}\cdot\pi(\boldsymbol{t})\,\mathrm{d}\boldsymbol{t}\right\}}{\int_\Omega e^{-kf(\boldsymbol{t})}\cdot\pi(\boldsymbol{t})\,\mathrm{d}\boldsymbol{t}} \\
&= -f(\boldsymbol{x}) - \frac{\int_\Omega(-f(\boldsymbol{t}))\cdot e^{-kf(\boldsymbol{t})}\cdot\pi(\boldsymbol{t})d\boldsymbol{t}}{\int_\Omega e^{-kf(\boldsymbol{t})}\cdot\pi(\boldsymbol{t})\,\mathrm{d}\boldsymbol{t}} \\
&= \mathbb{E}_{m_k}(f) - f(\boldsymbol{x}).
\end{aligned}$$

Hence,

$$\frac{\mathrm{d}}{\mathrm{d}k}m_k(\boldsymbol{x}) = m_k(\boldsymbol{x})\cdot\frac{\mathrm{d}}{\mathrm{d}k}\log m_k(\boldsymbol{x}) = m_k(\boldsymbol{x})\cdot(\mathbb{E}_{m_k}(f) - f(\boldsymbol{x})).$$

Then we have

$$\begin{aligned}
m^{(k+\Delta k)}(\boldsymbol{x}) &= m_k(\boldsymbol{x}) + \int_k^{k+\Delta k}\frac{\mathrm{d}}{\mathrm{d}\boldsymbol{v}}m^{(\boldsymbol{v})}(\boldsymbol{x})\,\mathrm{d}\boldsymbol{v} \\
&= m_k(\boldsymbol{x}) + \int_k^{k+\Delta k}m^{(\boldsymbol{v})}(\boldsymbol{x})\cdot\left[\mathbb{E}^{(\boldsymbol{v})}(f) - f(\boldsymbol{x})\right]\mathrm{d}\boldsymbol{v}.
\end{aligned}$$

Then there exist a $\xi \in (k, k + \Delta k)$ such that

$$\begin{aligned}
\frac{\mathbb{E}^{(k+\Delta k)}(f) - \mathbb{E}_{m_k}(f)}{\Delta k} &= \frac{1}{\Delta k}\int_\Omega f(\boldsymbol{x})\left(m^{(k+\Delta k)}(\boldsymbol{x}) - m_k(\boldsymbol{x})\right)\mathrm{d}\boldsymbol{x} \\
&= \frac{1}{\Delta k}\int_\Omega\int_k^{k+\Delta k}f(\boldsymbol{x})m^{(\boldsymbol{v})}(\boldsymbol{x})\left[\mathbb{E}^{(\boldsymbol{v})}(f) - f(\boldsymbol{x})\right]\mathrm{d}\boldsymbol{v}\,\mathrm{d}\boldsymbol{x} \\
&\stackrel{(R3)}{=} \frac{1}{\Delta k}\int_k^{k+\Delta k}\int_\Omega f(\boldsymbol{x})m^{(\boldsymbol{v})}(\boldsymbol{x})\left[\mathbb{E}^{(\boldsymbol{v})}(f) - f(\boldsymbol{x})\right]\mathrm{d}\boldsymbol{x}\,\mathrm{d}\boldsymbol{v} \\
&= \frac{1}{\Delta k}\int_k^{k+\Delta k}\left\{\left[\mathbb{E}^{(\boldsymbol{v})}(f)\right]^2 - \mathbb{E}^{(\boldsymbol{v})}(f^2)\right\}\mathrm{d}\boldsymbol{v} \\
&= \left[\mathbb{E}^{(\xi)}(f)\right]^2 - \mathbb{E}^{(\xi)}(f^2),
\end{aligned}$$

where the exchangeability of the integral is proved by Fubini Theorem in Remark 3. Hence, we have

$$\begin{aligned}
\frac{\mathrm{d}\mathbb{E}_{m_k}(f)}{\mathrm{d}k} &= \lim_{\Delta k\to 0}\frac{\mathbb{E}^{(k+\Delta k)}(f) - \mathbb{E}_{m_k}(f)}{\Delta k} \\
&= \{\mathbb{E}_{m_k}(f)\}^2 - \mathbb{E}_{(k)}(f^2) \\
&= -\int_\Omega(f(\boldsymbol{x}) - \mathbb{E}_{m_k}(f))^2\,m_k(\boldsymbol{x})\mathrm{d}\boldsymbol{x} \\
&= -\mathbb{V}ar^{(k)}(f) \leqslant 0,
\end{aligned}$$

where the equality holds only when $\mathbb{V}ar^{(k)}(f) = 0$, i.e., $f$ is a constant function on $\Omega$.

**Remark 3** (R3). *Define $h_k(t) = te^{-kt}$ for any $t \in \mathbb{R}$, then its first derivative is $h'_k(t) = (1 - kt)e^{-kt}$. For any $t < \frac{1}{k}$, $h'_k(t) > 0$, and thus $g(t)$ is increasing when $t < \frac{1}{k}$; for any $t \geqslant \frac{1}{k}$, $h'_k(t) \leqslant 0$, and thus $g(t)$ is non-increasing when $t \geqslant \frac{1}{k}$. Therefore, $h_k(t) \leqslant h_k(t = \frac{1}{k}) = \frac{1}{ke}$. Define $g_k(t) = t^2 e^{-kt}$, then the absolute value of $g_k(t)$ has its upper bound as $\frac{4}{k^2 e^2}$ using similar strategy. For any $k > 0$, the denominator of $\mathbb{E}^{(v)}(f)$ is a finite positive constant, where $0 < \int_\Omega e^{-kf(x)}\pi(x)\mathrm{d}x \triangleq \alpha_k \leqslant e^{-kf^*}\int_\Omega \pi(x)\mathrm{d}x \leqslant e^{-kf^*}$, and thus $\mathbb{E}^{(v)}(f)$ defined in the proof of Theorem 2 is bounded following:*

$$
\begin{aligned}
\mathbb{E}^{(v)}(f) &= \frac{\int_\Omega f(x)e^{-vf(x)}\pi(x)\mathrm{d}x}{\int_\Omega e^{-vf(x)}\pi(x)\mathrm{d}x} \\
&= \frac{\int_\Omega h_v(f(x))\pi(x)\mathrm{d}x}{\int_\Omega e^{-vf(x)}\pi(x)\mathrm{d}x} \\
&\leqslant \frac{\frac{1}{ve}\int_\Omega \pi(x)\mathrm{d}x}{\int_\Omega e^{-vf(x)}\pi(x)\mathrm{d}x} = \frac{1}{ve\alpha_v}
\end{aligned}
\tag{13}
$$

*Similarly, $\mathbb{E}^{(v)}(f^2)$ is also bounded by:*

$$
\begin{aligned}
\mathbb{E}^{(v)}(f^2) &= \frac{\int_\Omega f^2(x)e^{-vf(x)}\pi(x)\mathrm{d}x}{\int_\Omega e^{-vf(x)}\pi(x)\mathrm{d}x} \\
&= \frac{\int_\Omega g_v(f(x))\pi(x)\mathrm{d}x}{\int_\Omega e^{-vf(x)}\pi(x)\mathrm{d}x} \\
&\leqslant \frac{\frac{4}{v^2 e^2}\int_\Omega \pi(x)\mathrm{d}x}{\int_\Omega e^{-vf(x)}\pi(x)\mathrm{d}x} = \frac{4}{v^2 e^2 \alpha_v}
\end{aligned}
\tag{14}
$$

*Hence, given bounded $\mathbb{E}^{(v)}(f)$ and bounded $\mathbb{E}^{(v)}(f)$, the following order of integration is exchangeable by the Fubini theorem:*

$$
\int_\Omega \int_k^{k+\Delta k} \mathbb{E}^{(v)}(f)\mathrm{d}v \ \mathrm{d}x = \int_k^{k+\Delta k} \int_\Omega \mathbb{E}^{(v)}(f)\mathrm{d}x \ \mathrm{d}v,
$$

$$
\int_\Omega \int_k^{k+\Delta k} \mathbb{E}^{(v)}(f^2)\mathrm{d}v \ \mathrm{d}x = \int_k^{k+\Delta k} \int_\Omega \mathbb{E}^{(v)}(f^2)\mathrm{d}x \ \mathrm{d}v.
$$

$\square$

### A.3 PROOF FOR THEROEM 3

*Proof for Theroem 3.* First, we prove eq. (6).

For $\forall k > 0$, denote $c_k = \log\left\{\int_\Omega e^{-kf(t)}\pi(t) \, \mathrm{d}t\right\}$, then

$$
\begin{aligned}
\operatorname*{argmax}_{x \in \Omega} m_k(x) &= \operatorname*{argmax}_{x \in \Omega} \log m_k(x) \\
&= \operatorname*{argmax}_{x \in \Omega} \left(-kf(x) + \log \pi(x) - c_k\right) \\
&= \operatorname*{argmin}_{x \in \Omega} \left(f(x) - \frac{1}{k}\log \pi(x)\right),
\end{aligned}
$$

which indicates the maximizers of $m_k(x)$ are the minimizers of $\left(f(x) - \frac{1}{k}\log \pi(x)\right)$. Assume the upper bound for $\pi(x)$ is $B$, then for any $x_k^* \in \Omega_k$ and $\tilde{x} \in \Omega_*$, we have

$$
f(x_k^*) - \frac{1}{k}\log B \leqslant f(x_k^*) - \frac{1}{k}\log \pi(x_k^*) \leqslant f(\tilde{x}) - \frac{1}{k}\log \pi(\tilde{x}),
\tag{15}
$$

where the first inequality of eq. (15) holds for the bounded density, where $\pi(\boldsymbol{x}) \leqslant B, \forall \boldsymbol{x} \in \Omega$. The second inequality of eq. (15) lies in $\boldsymbol{x}_k^* \in \arg\max m_k(\boldsymbol{x})$. Furthermore, $\lim_{k\to\infty} f(\boldsymbol{x}_k^*) = f^*$ follows from

$$f^* \leqslant \liminf_{k\to\infty} f(\boldsymbol{x}_k^*) \leqslant \limsup_{k\to\infty} f(\boldsymbol{x}_k^*) \leqslant \limsup_{k\to\infty} \left\{ f(\tilde{\boldsymbol{x}}) - \frac{1}{k}\log\pi(\tilde{\boldsymbol{x}}) \right\} = f^*, \qquad (16)$$

where the three inequalities follow from the definition of $f^*$ ($\leqslant f(\boldsymbol{x})$, for $\forall \boldsymbol{x} \in \Omega$), the definition of limit inferior and limit superior, and the limit superior of eq. (15) respectively.

Next, we prove eq. (7). In the previous part, we have already proved $\lim_{k\to\infty} f(\boldsymbol{x}_k^*) = f^*$, i.e., for any $\epsilon > 0$, there exists large $K_0 > 0$, such that for $\forall k > K_0$, it holds that

$$f(\boldsymbol{x}_k^*) - f^* < \epsilon. \qquad (17)$$

Suppose there exists a large $K > K_0$, such that

$$\inf_{\tilde{\boldsymbol{x}}\in\Omega^*} \|\boldsymbol{x}_K^* - \tilde{\boldsymbol{x}}\| > \delta, \quad \text{for } \forall \tilde{\boldsymbol{x}} \in \Omega^*.$$

Then $f(\boldsymbol{x}_K^*) - f^* > \epsilon$ given the strong separability condition in Assumption 2, which is contradictory to eq. (17). Thus, we have completed the proof for Theorem 3 using the proof by contradiction technique. $\square$

## B  THEOREM AND DEFINITION

### B.1  PROPERTIES OF THE MINIMA DISTRIBUTION FUNCTION DEFINED ON NON-COMPACT SET

**Theorem 4.** *The nascent minima distribution function defined in Definition 1 satisfies:*

  *(i) For $\forall k \in \mathbb{R}$, $m_k(\boldsymbol{x})$ is a PDF on $\Omega$; especially when $k = 0$, $m^{(0)}(\boldsymbol{x}) = \pi(\boldsymbol{x})/\Pi(\Omega)$, where $\Pi(S) = \int_S \pi(\boldsymbol{x})\mathrm{d}\boldsymbol{x}$ is the probability measure of $S \subseteq \mathbb{R}^d$.*

  *(ii) If $\nabla\pi$ and $\nabla f : \mathbb{R}^d \to \mathbb{R}^d$ are continuous real functions on each dimension, then*

$$\nabla \log m_k(x) = -k\nabla f(x) + \frac{\nabla\pi(x)}{\pi(x)}.$$

  *(ii) For every $k \in \mathbb{R}$, it holds that*

$$\frac{\mathrm{d}}{\mathrm{d}k} \log m_k(\boldsymbol{x}) = \mathbb{E}_{m_k}(f) - f(\boldsymbol{x}),$$

  *where $\mathbb{E}_{m_k}(f) = \int_\Omega f(\boldsymbol{x})m_k(\boldsymbol{x})\mathrm{d}\boldsymbol{x}$.*

  *(iii) For any $\boldsymbol{x} \in \Omega$, consider the sequence $m_k(\boldsymbol{x})$, where $k > 0$ and is going to infinity:*

  *(a) If $\Omega^*$ has zero probability measure, i.e., $\Pi(\Omega^*) \triangleq \int_{\Omega^*} \pi(\boldsymbol{x})\mathrm{d}\boldsymbol{x} = 0$, then*

$$m_\infty(\boldsymbol{x}) = \lim_{k\to\infty} m_k(\boldsymbol{x}) = \left\{ \begin{array}{ll} 0, & \boldsymbol{x} \notin \Omega^*; \\ \infty, & \boldsymbol{x} \in \Omega^*. \end{array} \right.$$

  *(b) If $\Omega^*$ has nonzero probability measure, i.e., $\Pi(\Omega^*) > 0$, then*

$$\lim_{k\to\infty} m_k(\boldsymbol{x}) = \left\{ \begin{array}{ll} 0, & \boldsymbol{x} \notin \Omega^*; \\ \frac{\pi(\boldsymbol{x})}{\Pi(\Omega^*)}, & \boldsymbol{x} \in \Omega^*. \end{array} \right.$$

  *(iv) Define $\Omega_k$ as the set of maximizers of $m_k(\boldsymbol{x})$ on $\boldsymbol{x} \in \Omega$, if $\pi(\boldsymbol{x})$ is bounded, then for any $k > 0$, the sequence of $\boldsymbol{x}_k^* \in \Omega_k$ satisfies the following:*

  *(a) $\lim_{k\to\infty} f(\boldsymbol{x}_k^*) = 0(= f^*)$.*

  *(b) $\pi(\boldsymbol{x}_k^*) \geqslant \pi^* \triangleq \max_{\boldsymbol{x}^*\in\Omega^*} \pi(\boldsymbol{x}^*)$.*

(c) *The sequence $\{\pi(\boldsymbol{x}_k^*)\}_{k=1,2,\ldots}$ is non-increasing and converges to a limit, where $\lim_{k\to\infty}\pi(\boldsymbol{x}_k^*) = \liminf_{k\to\infty}\pi(\boldsymbol{x}_k^*) \overset{\Delta}{=} \pi_0$.*

*Proof.* Clearly, (i) follows from the definition of $m_k(\boldsymbol{x})$ in Definition 1.

- For every $k \in \mathbb{R}$, (ii) follows from

$$
\begin{aligned}
\frac{\mathrm{d}}{\mathrm{d}k}\log m_k(\boldsymbol{x}) &= \frac{\mathrm{d}}{\mathrm{d}k}\left(-kf(\boldsymbol{x}) + \log\pi(\boldsymbol{x}) - \log\left\{\int_\Omega e^{-kf(t)}\cdot\pi(t)dt\right\}\right) \\
&= -f(\boldsymbol{x}) - \frac{\frac{\mathrm{d}}{\mathrm{d}k}\left\{\int_\Omega e^{-kf(t)}\cdot\pi(t)dt\right\}}{\int_\Omega e^{-kf(s)}\cdot\pi(s)ds} \\
&= -f(\boldsymbol{x}) - \frac{\int_\Omega(-f(t))\cdot e^{-kf(t)}\cdot\pi(t)dt}{\int_\Omega e^{-kf(s)}\cdot\pi(s)ds} \\
&= \mathbb{E}_{m_k}(f) - f(\boldsymbol{x}),
\end{aligned}
$$

where $\frac{\mathrm{d}}{\mathrm{d}k}\left\{\int_\Omega e^{-kf(t)}\cdot\pi(t)dt\right\} = \int_\Omega\frac{\mathrm{d}}{\mathrm{d}k}\left\{e^{-kf(t)}\cdot\pi(t)\right\}dt$ follows from the Tonelli theorem for the exchangeable order of the integral and derivatives, given that the function $\left\{e^{-kf(t)}\cdot\pi(t)\right\}$ is non-negative.

- For (iii) notice that by Remark 1,

$$
\int_\Omega e^{-kf(\boldsymbol{x})}\pi(\boldsymbol{x})\mathrm{d}\boldsymbol{x} = \int_{\Omega^*}\pi(\boldsymbol{x})\mathrm{d}\boldsymbol{x} + \int_{\Omega\setminus\Omega^*}e^{-kf(\boldsymbol{x})}\pi(\boldsymbol{x})\mathrm{d}\boldsymbol{x},
$$

since $f(\boldsymbol{x}) = 0$ for $\forall\boldsymbol{x}\in\Omega^*$. In addition, since $\left\{e^{-kf(\boldsymbol{x})}\pi(\boldsymbol{x})\right\}$ is monotonely decreasing as $k$ increases and $\lim_{k\to\infty0}\left\{e^{-kf(\boldsymbol{x})}\pi(\boldsymbol{x})\right\} = 0$ due to $f(\boldsymbol{x}) > 0$ for any $\boldsymbol{x}\notin\Omega^*$, then

$$
\lim_{k\to\infty}\int_\Omega e^{-kf(\boldsymbol{x})}\pi(\boldsymbol{x})\mathrm{d}\boldsymbol{x} = \int_{\Omega^*}\pi(\boldsymbol{x})\mathrm{d}\boldsymbol{x} + \lim_{k\to\infty}\int_{\Omega\setminus\Omega^*}e^{-kf(\boldsymbol{x})}\pi(\boldsymbol{x})\mathrm{d}\boldsymbol{x} = \Pi(\Omega^*),
$$

which follows from the monotone convergence theorem.

(a) Hence, for any $\boldsymbol{x}\in\Omega^*$,

$$
m_\infty(\boldsymbol{x}) = \lim_{k\to\infty}m_k(\boldsymbol{x}) = \lim_{k\to\infty}\frac{e^{-kf(\boldsymbol{x})}\cdot\pi(\boldsymbol{x})}{\int_\Omega e^{-kf(t)}\cdot\pi(t)dt} = \begin{cases} \infty, & \text{if } \Pi(\Omega^*) = 0; \\ \frac{\pi(\boldsymbol{x})}{\Pi(\Omega^*)}, & \text{if } \Pi(\Omega^*) \neq 0. \end{cases}
$$

(b) For any $\boldsymbol{x}'\notin\Omega^*$, it follows from the continuity of $f$ that there exists a set $\Omega_{\boldsymbol{x}'}$ such that $f(t) < f(\boldsymbol{x}')$ for any $t\in\Omega_{\boldsymbol{x}'}$, hence,

$$
\begin{aligned}
m_k(\boldsymbol{x}') &= \frac{e^{-kf(\boldsymbol{x}')}\cdot\pi(\boldsymbol{x}')}{\int_{\Omega_{\boldsymbol{x}'}}e^{-kf(t)}\cdot\pi(t)dt + \int_{\Omega\setminus\Omega_{\boldsymbol{x}'}}e^{-kf(t)}\cdot\pi(t)dt} \\
&\leqslant \frac{\pi(\boldsymbol{x}')}{\int_{\Omega_{\boldsymbol{x}'}}(e^{-f(t)}/e^{-f(\boldsymbol{x}')})^k\cdot\pi(t)dt},
\end{aligned}
$$

since $e^{-f(t)}/e^{-f(\boldsymbol{x}')} > 1$ for any $t \in \Omega_{\boldsymbol{x}'}$ and $\pi(t) > 0$, the limit of $\int_{\Omega_{\boldsymbol{x}'}}(e^{-f(t)}/e^{-f(\boldsymbol{x}')})^k\cdot\pi(t)dt$ tends to $\infty$ as $k\to\infty$; thus, it holds that

$$
\lim_{k\to\infty}m_k(\boldsymbol{x}') = 0, \forall\boldsymbol{x}'\notin\Omega^*.
$$

- (iv) describes the properties of the maximizers of $m_k(\boldsymbol{x})$. For $\forall k > 0$, denote $c_k = \log\left\{\int_\Omega e^{-kf(t)}\pi(t)dt\right\}$, then

$$
\begin{aligned}
\operatorname*{argmax}_{\boldsymbol{x}\in\Omega} m_k(\boldsymbol{x}) &= \operatorname*{argmax}_{\boldsymbol{x}\in\Omega} \log m_k(\boldsymbol{x}) \\
&= \operatorname*{argmax}_{\boldsymbol{x}\in\Omega} \left(-kf(\boldsymbol{x}) + \log\pi(\boldsymbol{x}) - c_k\right) \\
&= \operatorname*{argmin}_{\boldsymbol{x}\in\Omega} \left(f(\boldsymbol{x}) - \frac{1}{k}\log\pi(\boldsymbol{x})\right) \\
&= \operatorname*{argmin}_{\boldsymbol{x}\in\Omega} g_k(\boldsymbol{x}),
\end{aligned}
$$

which indicates the maximizers of $m_k(\boldsymbol{x})$ are the minimizers of $g_k(\boldsymbol{x})$, i.e., $\Omega_k = \{\boldsymbol{x}' : m_k(\boldsymbol{x}') \geqslant m_k(\boldsymbol{x}), \forall \boldsymbol{x} \in \Omega\} = \{\boldsymbol{x}'' : g_k(\boldsymbol{x}'') \leqslant g_k(\boldsymbol{x}), \forall \boldsymbol{x} \in \Omega\}$.

(a) For any $\boldsymbol{x}_k^* \in \Omega_k$ and $\boldsymbol{x}^* \in \Omega^*$, we have

$$
f(\boldsymbol{x}_k^*) - \frac{1}{k}\log B \leqslant f(\boldsymbol{x}_k^*) - \frac{1}{k}\log\pi(\boldsymbol{x}_k^*) \leqslant f(\boldsymbol{x}^*) - \frac{1}{k}\log\pi(\boldsymbol{x}^*),
$$

where the first inequality lies in $\pi(\boldsymbol{x}) \leqslant B, \forall \boldsymbol{x} \in \Omega$ and the second inequality lies in $\boldsymbol{x}_k^* \in \arg\max m_k(\boldsymbol{x})$. Furtherly, $\lim_{k\to\infty} f(\boldsymbol{x}_k^*) = f^*$ follows from

$$
f^* \leqslant \liminf_{k\to\infty} f(\boldsymbol{x}_k^*) \leqslant \limsup_{k\to\infty} f(\boldsymbol{x}_k^*) \leqslant \limsup_{k\to\infty}\left\{f(\boldsymbol{x}^*) - \frac{1}{k}\log\pi(\boldsymbol{x}^*)\right\} = f^*.
$$

(b) For any $\boldsymbol{x}^* \in \Omega^*$, and $\boldsymbol{x}_k^* \in \Omega_k$, we have $\pi(\boldsymbol{x}_k^*) \geqslant \pi^* = \max_{\boldsymbol{x}\in\Omega^*}\pi(\boldsymbol{x}^*)$ for $k > 0$, which follows from:

$$
f(\boldsymbol{x}_k^*) - \frac{1}{k}\log\pi(\boldsymbol{x}_k^*) \leqslant f(\boldsymbol{x}^*) - \frac{1}{k}\log\pi(\boldsymbol{x}^*) \leqslant f(\boldsymbol{x}_k^*) - \frac{1}{k}\log\pi(\boldsymbol{x}^*).
$$

(c) The monotonicity of the sequence $\{\pi(\boldsymbol{x}_k^*)\}_{k=1,2,\dots}$ follows from:

$$
\begin{aligned}
& f(\boldsymbol{x}_{k+1}^*) - \frac{1}{k+1}\log\pi(\boldsymbol{x}_{k+1}^*) \leqslant f(\boldsymbol{x}_k^*) - \frac{1}{k+1}\log\pi(\boldsymbol{x}_k^*) \\
\iff & f(\boldsymbol{x}_{k+1}^*) - \frac{1}{k+1}\log\pi(\boldsymbol{x}_{k+1}^*) \leqslant f(\boldsymbol{x}_k^*) - \frac{1}{k}\log\pi(\boldsymbol{x}_k^*) + \frac{1}{k(k+1)}\log\pi(\boldsymbol{x}_k^*) \\
\iff & f(\boldsymbol{x}_{k+1}^*) - \frac{1}{k+1}\log\pi(\boldsymbol{x}_{k+1}^*) \leqslant f(\boldsymbol{x}_{k+1}^*) - \frac{1}{k}\log\pi(\boldsymbol{x}_{k+1}^*) + \frac{1}{k(k+1)}\log\pi(\boldsymbol{x}_k^*) \\
& \iff \log\pi(\boldsymbol{x}_{k+1}^*) \leqslant \log\pi(\boldsymbol{x}_k^*)
\end{aligned}
$$

$\square$

## B.2 EXAMPLE ON UNBOUNDED SET

This section provides an example of constructing $\pi(x)$ for an unbounded set $\Omega$, such as $\mathbb{R}^d$. To address the challenge of an unbounded domain, one effective approach is to employ a generalized normal type distribution for the prior density. This choice provides an approximation to uniformity within a specified range while ensuring that the distribution maintains well-behaved properties, making it suitable for our optimization framework in unbounded scenarios.

**Definition 3.** *Assume that there exists a large $M > 0$ such that $\Omega_* \subseteq [-M, M]^d$. Define a new density function as*

$$
\pi(x) = \begin{cases} \frac{w}{(2M)^d}, & x \in [-M, M]^d; \\ (1-w)p(x), & x \notin [-M, M]^d, \end{cases} \quad \text{for } \forall w \in (0,1), \tag{18}
$$

*where $w$ is a weight parameter between $(0,1)$, and $p(x)$ is any arbitrary density function such that $\int_{|x|>M} p(x)\mathrm{d}x = 1$.*

---

**Algorithm 2:** Latent Slice Sampler for ProGO (LSS-ProGO)

---

**Input:** Target probability distribution $\pi : \Omega \to [0, 1]$, sample size $N$, initialization $\boldsymbol{x}^{(0)}$, burn-in period $n_b$

1   Initialize $\boldsymbol{x} = \boldsymbol{x}^{(0)}$, $w^{(0)} \sim U(0, \pi(\boldsymbol{x}^{(0)}))$, $\boldsymbol{s}_j^{(0)} \sim \mathrm{Gamma}(2, \beta)$, for $j = 1, \cdots, d$, $\boldsymbol{l}^{(0)} \sim U(\boldsymbol{x}^{(0)} - \boldsymbol{s}^{(0)}/2, \boldsymbol{x}^{(0)} + \boldsymbol{s}^{(0)}/2)$

2   **for** iteration $t \in \{1, 2, \cdots, N + n_b\}$ **do**

3      $\boldsymbol{a} \leftarrow \boldsymbol{l}^{(t-1)} - \boldsymbol{s}^{(t-1)}/2$

4      $\boldsymbol{b} \leftarrow \boldsymbol{l}^{(t-1)} + \boldsymbol{s}^{(t-1)}/2$

5      **while** $\boldsymbol{x} \notin \{\boldsymbol{x} : \pi(\boldsymbol{x}) > w^{(t-1)}\}$ **do**

6          **for** dimension $j \in \{1, \cdots, d\}$ **do**

7              **if** $\boldsymbol{x}_j < \boldsymbol{x}_j^{(t-1)}$ **then**

8                  $\boldsymbol{a}_j \leftarrow \max(\boldsymbol{a}_j, \boldsymbol{x}_j)$

9              **else**

10                  $\boldsymbol{b}_j \leftarrow \min(\boldsymbol{b}_j, \boldsymbol{x}_j)$

11              **end**

12          **end**

13          $\boldsymbol{x} \sim U(\boldsymbol{a}, \boldsymbol{b})$

14      **end**

15      $\boldsymbol{x}^{(t)} \leftarrow \boldsymbol{x}$

16      $w^{(t)} \sim U(0, \pi(\boldsymbol{x}^{(t)}))$

17      $\boldsymbol{s}^{(t)} \sim e^\beta + 2|\boldsymbol{l}^{(t)} - \boldsymbol{x}^{(t)}|$

18      $\boldsymbol{l}^{(t)} \sim U(\boldsymbol{x}^{(t)} - \boldsymbol{s}^{(t)}/2, \boldsymbol{x}^{(t)} + \boldsymbol{s}^{(t)}/2)$

19   **end**

**Output:** Samples $\{\boldsymbol{x}^{(n_b+1)}, \cdots, \boldsymbol{x}^{(n_b+N)}\}$ from target probability distribution $\pi$

---

## C   ALGORITHM

The family of slice samplers (SS) is well-known for their geometric rates of convergence properties. For instance, Roberts and Rosenthal (1999) demonstrated the near-geometric ergodic behavior of SS, and Mira and Tierney (2002) provided conditions for SS to achieve uniform ergodicity. As an extension of SS, the detailed algorithm of latent slice sampler (LSS) and its practical convergence have been discussed in Li and Walker (2023). Our method of approximating the minima is based on the stochastic sampling from the minima distribution $m_k(x)$ using LSS. Hence, obtaining non-asymptotic bounds is a non-trivial task, which we will explore in future research.

It it noteworthy that there is no need to compute the denominator in the eq. (2). This is because $m_k(x) \propto e^{-kf(x)} \cdot \pi(x)$, and the integral serves as a constant. As it's well-known within the MCMC literature, the normalizing constant of the density is not required to generate samples from the path of the Markov chain (by using the LSS). To clarify, for any black-box target function, all that are required are MCMC samples generated from the kennel of the density $e^{-kf(x)} \cdot \pi(x)$ to approximate the minima distribution in equation (2). Hence, there's no need to compute the denominator to implement the LSS.

Determining the appropriate value of $k$ depends on the specific characteristics of the target function, including its value and dimension. In Algorithm 1, employing a single large $k$ within the exponential can result in numerical instability in the sampling algorithm. Our goal is to find $k$ sufficiently large enough to make LSS numerically stable until the convergence of the minima is numerically achieved. On the other hand, the latent slice sampler demonstrates a geometric rate of convergence even when with smaller values of $k$. Despite no direct interaction between iterations, the algorithm can achieve the desired results effectively. This justifies our approach of conducting a sequential search for the optimal k, considering both practical and theoretical considerations. Hence, throughout our sequential search, we explored a wide range of $k$ values, covering from $k = 5$ to $k = 5e^{200}$. As our method is shown to converge theoretically to the minima, our goal here is to approximate the minima by slowly increasing the $k$ (maintaining the stability of the LSS) until a numerical conference is observed.

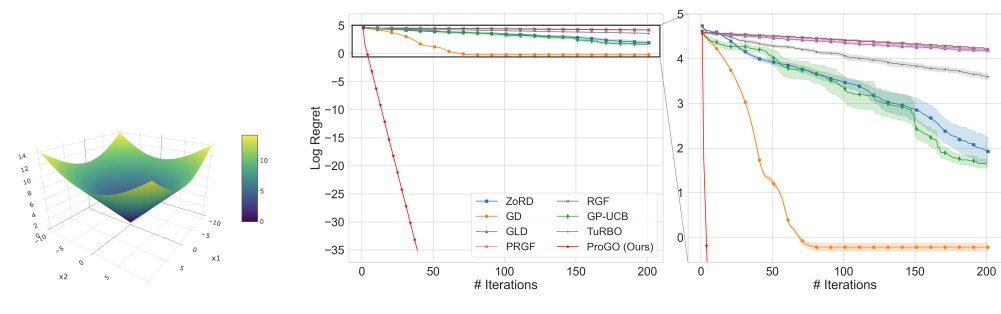

(a) Surface of $\mathcal{L}$-2 norm ($d = 2$)    (b) Results on $\mathcal{L}$-2 norm ($d = 40$)

Figure 4: (a) Visualization of the two-norm function for $d = 2$. (b) Evaluation of ProGO and competing methods applied to the Levy function in a high-dimensional setting with $d = 1000$. The $x$-axis represents the iteration count, while the $y$-axis denotes the average log-scaled regret. Each curve shows the mean $\pm$ standard error across ten independent runs.

# D EXPERIMENTS

## D.1 CONVEX EXAMPLES

In addition to the highly non-convex examples, we conducted additional experiments to evaluate ProGO's performance on convex target functions. Two-norm function, also known as the Euclidean norm, is an example taken from Mazumder et al. (2019), which focuses on multivariate convex regression and provides various types of convex functions.

$$f(\mathbf{x}) = \|\mathbf{x}\|_2^2, \tag{19}$$

This function is characterized as a smooth and convex function with a central global minimum at $f^* = 0$ and $\boldsymbol{x}^* = (0, \cdots, 0)^\top$. Our empirical evaluations (as presented in Table 3) cover

Table 3: Comparison of performance on the two-norm function for dimensions $d = 40, 1000$: Each optimization method undergoes ten independent runs. Accuracy is quantified using the average function log regret ($r_f$) and the average minima log regret ($r_m$) across these ten runs. Computational efficiency is represented by the average runtime ($t$) across all ten runs, measured in seconds.

| Method | $d = 40$ | | | $d = 1000$ | | |
|---|---|---|---|---|---|---|
| | $r_f$ | $r_m$ | $t$ (s) | $r_f$ | $r_m$ | $t$ (s) |
| ZoRD | 1.93 | 1.61 | 7062.84 | 5.39 | 1.63 | 3185.37 |
| GD | -0.21 | 1.61 | 1.15 | **0.95** | 1.61 | 1.25 |
| GLD | 4.18 | 1.66 | 3.76 | 6.26 | 1.74 | 5.81 |
| PRGF | 4.19 | 1.67 | 0.58 | 6.25 | 1.75 | 0.69 |
| RGF | 4.22 | 1.66 | 0.56 | 6.24 | 1.74 | 0.74 |
| GP-UCB | 1.65 | 1.63 | 1878.19 | 6.19 | 1.74 | 11182.87 |
| TuRBO | 3.60 | 1.61 | 805.23 | 6.19 | 1.73 | 2555.74 |
| **ProGO** | **-35.15** | **-35.15** | 102.02 | 1.01 | **1.01** | 1959.02 |

dimensions $d = 40$ and $1000$. In the case of $d = 40$, ProGO consistently outperforms all other methods, achieving significantly lower function log regret and lower minima log regret. However, as dimensionality increases to $d = 1000$, the advantage of using direct gradient information becomes evident, with GD obtaining the lowest function log regret. Nevertheless, ProGO still maintains the lowest minima regret in this high-dimensional scenario, surpassing all other algorithms except GD. Furthermore, Figures 4 and 7 offer visual evidence of ProGO's geometric rate of convergence.

## D.2 EXPERIMENTAL SETTINGS

Consistent with the experimental parameters adopted by ZoRD (Shu et al., 2022), we designate the domains of the Ackley, Levy, and two-norm functions as $[-20, 20]^d$, $[-7.5, 7.5]^d$, $[-30, 30]^d$ respectively, and chose the stopping criteria as 200 iterations for each of the methods. For ProGO, the parameter for LSS is sample size as $N = 200$ and burn-in period $n_b = 20$. The configurations are uniformly applied across the RGF, PRGF, GD, and ZoRD algorithms to ensure fair comparisons, where the Adam optimizer (Kingma and Ba, 2015) is utilized with a fixed learning rate of 0.1 and exponential decay rates of 0.9 and 0.999. It should be noted that while ProGO is implemented in the R environment, other algorithms are executed in Python. Although this discrepancy may affect runtime comparisons, it does not influence accuracy results.

## D.3 SUPPLEMENTARY RESULTS

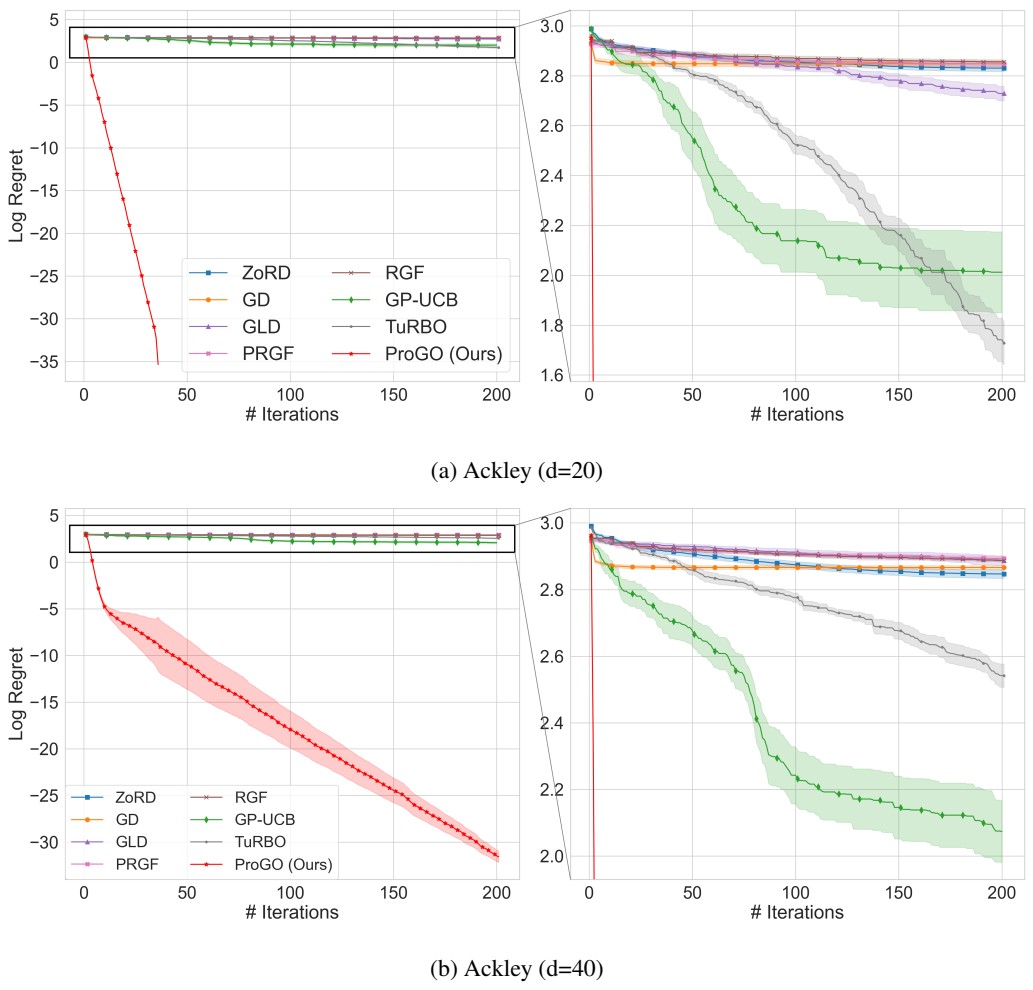

(a) Ackley (d=20)

(b) Ackley (d=40)

Figure 5: Evaluation of ProGO and competing methods applied to the Ackley function with $d = 20, 40$. The $x$-axis represents the iteration count, while the $y$-axis denotes the average log-scaled regret. Each curve shows the mean $\pm$ standard error across ten independent runs.

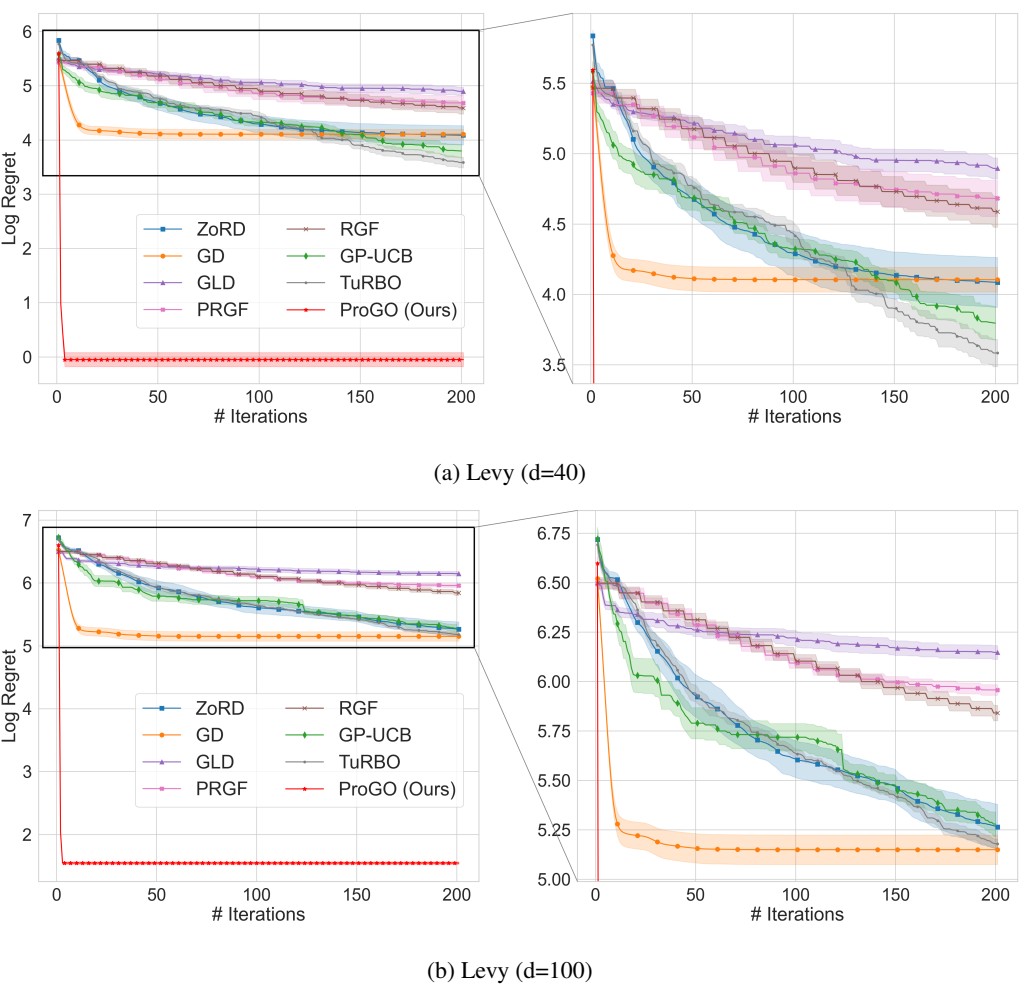

(a) Levy (d=40)

(b) Levy (d=100)

Figure 6: Evaluation of ProGO and competing methods applied to the Ackley function with $d = 40, 100$. The $x$-axis represents the iteration count, while the $y$-axis denotes the average log-scaled regret. Each curve shows the mean $\pm$ standard error across ten independent runs.

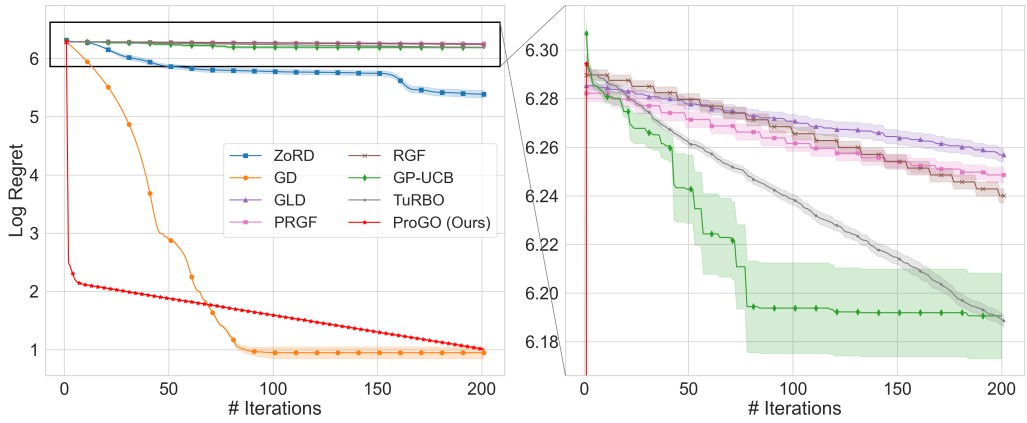

Figure 7: Evaluation of ProGO and competing methods applied to the two-norm function in a high-dimensional setting with $d = 1000$. The $x$-axis represents the iteration count, while the $y$-axis denotes the average log-scaled regret. Each curve shows the mean $\pm$ standard error across ten independent runs.

