# OpenReview forum: "ProGO: Probabilistic Global Optimizer"
_ICLR.cc/2024/Conference — Submitted to ICLR 2024_

### Official Review · Reviewer_WwVi · 2023-10-29

**Soundness:** 2 fair
**Presentation:** 3 good
**Contribution:** 1 poor
**Rating:** 3
**Confidence:** 3

**Summary:**

The authors describe an optimization algorithm for continuous functions $f:\mathbb{R^d}\to \mathbb{R}$ that draws random samples from the search space using a Latent slice sampler and the probability density from which the samples are drawn is the product of a prior $\pi(x)$ and the exponentiated objective functions $\exp(-k\cdot f(x))$, i.e. an energy based model.

The authors prove basic properties of the distribution and perform experiments against a range of comparable methods.

**Strengths:**

- well written, clear and easy to follow
- the idea of using the objective function as energy in an energy based model together with an MCMC algorithm and varying temperature is is well justified (and unfortunately has been done extensively before)

**Weaknesses:**

- using the objective as energy in an energy boltzmann distribution (also known as energy based models) for MCMC sampling is an old well established family of methods known as simulated annealing. This paper does not acknowledge simulated annealing nor cite any simulated annealing work (the prior work of Luo 2018 cites one paper for simulated annealing when discussing differential evolution). I feel the proposed framework is not novel. In the simulated annealing literature, $1/k$ is referred to as temperature $T$ and increasing $k$ or reducing $T$ over time is the “cooling schedule”, and as the algorithm "cools down", the sampler converges closer and closer to the minima (exactly as described by Theorem 1).

- it is not clear if the methods are using the same number of objective function evaluations. If I understand Algorithm 1 correctly, the objective function is evaluated in line 5, in the condition of the while loop, hence the while loops and thus the number of objective function calls is not deterministic for each iteration of the algorithm. Comparing algorithms by number of iterations when algorithms call the objective a different number of times each is not a fair comparison.

- the proofs are standard results, rewriting the minima distribution in exponential family form $m_k(x) = exp([-k, 1] \cdot [f(x), \log(pi(x))]) / Z$, Theorem 2 is a standard result for all exponential family distribution, the gradients with respect to the natural parameters yield moments of the distribution.

- as the temperature of an energy based model is reduced, the distribution tends to delta function around the maximum is well known. In many generative models, increasing the “temperature” in sampling is synonymous with increasing variety in generated outputs, and decreasing temperature leads to deterministically generating maximum likelihood outputs.

- in my view the framework of ProGO is not new. The use of this particular MCMC Latent Slice Sampler within an simulated annealing algorithm may be new. If so, Standard Metropolis Hasting with a Gaussian proposal and other MCMC methods should be baselines. Even so, a change of MCMC sampler is not sufficient for publication.

I enjoyed the paper and the writing, however, unfortunately I cannot recommend for acceptance.

**Questions:**

- do all algorithms evaluate $f(x)$ exactly the same number of times per iteration? the results should plot total numebr of function $f(x)$ calls on the x axis.

---

> ### Author Response · Authors · 2023-11-22
>
> > Weakness: in my view the framework of ProGO is not new. The use of this particular MCMC Latent Slice Sampler within an simulated annealing algorithm may be new. If so, Standard Metropolis Hasting with a Gaussian proposal and other MCMC methods should be baselines. Even so, a change of MCMC sampler is not sufficient for publication.
>
> We respectfully disagree with the view. Yes, there are similarities between our approach and simulated annealing in the sense that they both use sampling methods. Simulated annealing uses Metropolis-Hastings (MH) random walk while we utilize the Latent Slice Sampler (LSS), which is an alternative better than MH. However, it is well known in the literature, Random Walk-Metropolis Hastings (RW-MH) methods, particularly those based on Gaussian perturbations, are known to have slower rates of convergence compared to slice samplers, which enjoy a geometric rate of convergence.
>
> To the best of our knowledge, the theoretical convergence results for target functions, which are assumed only to be continuous (without any smoothness conditions) and possibly non-convex, have not been established when the domain is unbounded. Furthermore, our theoretical extension from compact to non-compact sets is nontrivial given the relaxation of conditions. This is our first novelty.
>
> Second, to the best of our knowledge, the use of LSS for optimization has not been available in the existing literature.
>
> We have not come across any relevant references, and we would be grateful if the reviewer could provide any if applicable.
>
>
>
>
> > Question: Do all algorithms evaluate exactly the same number of times per iteration?
>
> We acknowledge that the function evaluations are not once for each iteration of ProGO, and the count of function evaluations per iteration may differ among various algorithms due to variations in their optimization strategies as well. Hence, we employ runtime as a standardized metric to assess and compare different methods impartially.

---

### Official Review · Reviewer_jr1B · 2023-10-31

**Soundness:** 4 excellent
**Presentation:** 3 good
**Contribution:** 3 good
**Rating:** 6
**Confidence:** 4

**Summary:**

This paper considers the problem of finding global minimums of non-convex functions. It proposed a new algorithm that does not require the computation of gradients, but rather it depends on sampling from a sequence of distributions $m_k(x)$ that is induced by the objective function $f$. The authors showed that the maximizer of each distribution $x_k^* = \mathrm{argmax} (m_k(x))$ converges asymptotically to a global minimum, under some separation conditions. In addition, $f(x_k)$ converges to $f^*$ even when the separation condition does not hold. The authors also performed experiments on some non-convex functions and showed that the proposed algorithm converges much faster than other algorithms, like gradient descent etc.

**Strengths:**

1. This paper is thoeretically interesting, in that it proposed a new type of optimization algorithms and constructs its (asymptotic) convergence theory. This algorithm provably converges to the global minimum of the objective. Interestingly, they showed that $\int f(x)m_k(x) dx \downarrow f^*$, where $m_k(x)$ is a probability measure constructed using $f$. Further, when $x_k^* = \mathrm{argmax} (m_k(x))$, it holds that $x^*_k \to x^*$ in $\ell_2$ distance.
2. This paper is written clearly and provides useful explanation and intuition.

**Weaknesses:**

1. The authors only provides convergence property when the iteration $k$ goes to infinity. It seems promising that the new method out-performs classic algorithms on some functions, but what about the worst-case upper bound? It would be more interesting (and practical) if we have some non-asymptotic results like how many iterations we need to approximate an global minimizer within error $\epsilon$. If, in the worst case, the algorithm needs $(1/\epsilon)^d$ calls of function value oracle to find a global min of a (say, lipschitz) function, then it is no better than a brute force search.

**Questions:**

1. How many samples does the LSS-ProGo algorithm need to approximate the distribution $m_k(x)$? Especially, in high-dimension scenerios? How accurate this approximation needs to be?
2. I am wondering what would happen if we use Gaussian as $\pi(x)$ instead of uniform distribution. Will that bring us better results?
3. In addition, when the set $\Omega$ is unbounded, like $R^d$, how to construct a uniform distribution on this set $\Omega$?

---

> ### Author Response · Authors · 2023-11-22
> **Response to Reviewer jr1B**
>
> > W1: “The authors only provides convergence property when the iteration
>  goes to infinity. It seems promising that the new method out-performs classic algorithms on some functions, but what about the worst-case upper bound? It would be more interesting (and practical) if we have some non-asymptotic results like how many iterations we need to approximate an global minimizer within error. If, in the worst case, the algorithm needs calls of function value oracle to find a global min of a (say, lipschitz) function, then it is no better than a brute force search.”
>
> Our method of approximating the minima employs stochastic latent slice sampling from the minima distribution $m_k(x)$. While obtaining non-asymptotic bounds is a challenging task we aim to tackle in future research, it's important to highlight that our method fundamentally differs from traditional sampling methods, utilizing the theoretical property of the minima distribution and the practical efficiency of the LSS technique. In contrast, brute force search explores the entire space and is unsuitable for high dimensions. Thus, our algorithm's robust performance in low- and high-dimensional cases demonstrates its superiority over brute-force search.
>
> We appreciate the insights to conduct further explorations to validate our algorithm's performance for different scenario. In response, we have included additional examples such as convex optimization in Appendix B and plan to explore other applications in our future research.
>
> > Q1: “How many samples does the LSS-ProGo algorithm need to approximate the distribution? Especially, in high-dimension scenerios? How accurate this approximation needs to be?”
>
> The required Monte Carlo (MC) sample size to achieve a desired level of precision in approximating the minimum distribution from MC samples can be grounded in fundamental statistical principles, including Central Limit Theory (CLT), Strong Law of Large Numbers (SLLN), Strong Law of Iterated Logarithm (SLIL), and Hoeffding's Inequality (e.g., [1], [2], [3]). For example, when adopting a CLT-based approach to ensure an error no greater than $\epsilon$, it is essential to have a sample size of at least $N \geq 4 \hat{\sigma}^2$, where $\hat{\sigma}^2 = \frac{1}{N-1} \sum_{i=1}^N(f(x_i)-f(x_i^*))^2$. It's noteworthy that the stochastic method's MC sample size is dimensional-invariant, while deterministic methods may experience the curse of dimensionality.
>
> [1] Caflisch, Russel E. "Monte carlo and quasi-monte carlo methods." Acta numerica 7 (1998): 1-49.
> [2] Zaremba, Stanislaw K. "The mathematical basis of Monte Carlo and quasi-Monte Carlo methods." SIAM review 10.3 (1968): 303-314.
> [3] Reich, Brian J., and Sujit K. Ghosh. Bayesian statistical methods. CRC Press, 2019.
>
> > Q2: I am wondering what would happen if we use Gaussian as instead of uniform distribution. Will that bring us better results?
>
>
> Thank you for this insightful suggestion. Using a Gaussian distribution as an alternative to a uniform one introduces additional prior information, involving the specification of the mean and covariance. If the Gaussian prior's mean closely aligns with the global optimum and the covariance is appropriately set, it has the potential to expedite convergence. In response, we appreciate the idea of exploring Gaussian priors in our framework. We will consider conducting further experiments in future research to assess the impact of Gaussian priors on ProGO's performance.
>
>
> > Q3: In addition, when the set is unbounded, like how to construct a uniform distribution on this set?
>
> Good question, we have included a description of such an approach in Appendix B.2 to provide clarity on this matter. We could use a generalized normal type distribution for the prior density which provides an approximation to uniform with tails vanishing exponentially outside a desired range.

---

### Official Review · Reviewer_dC7R · 2023-10-31

**Soundness:** 3 good
**Presentation:** 3 good
**Contribution:** 2 fair
**Rating:** 3
**Confidence:** 4

**Summary:**

The authors present a new probabilistic global optimization algorithm called ProGO, which relies on the theory of nascent minima distribution by Luo (2018) and the latent slice sampler. This method departs from traditional gradient-based approaches by ensuring reliable convergence to global optima without the need for gradient information, while still maintaining computational efficiency. They extend Lou’s framework to accommodate noncompact sets and demonstrate its global convergence. The ProGO algorithm is then developed based on this extended framework, and it incorporates a latent slice sampler to improve computational efficiency.

**Strengths:**

1)	Propose a novel derivative-free optimization algorithm for global optimization with convergence guarantee.

2)	Extend the framework proposed in Luo (2018) to non-compact constraint sets and analyze its global convergence.

3)	Provide some promising experimental results.

**Weaknesses:**

1)	In order to apply the proposed algorithm, the optimal function value $f^*$ is supposed to be known. The assumption is not true for many practical applications.

2)	The algorithm is only tested on two test instances, which can be efficiently solved by a number of existing derivative-free optimization algorithms. More experiments are expected to convince the performance benefit of the algorithm, for example, see “N. Hansen, A. Auger, R. Ros, O. Mersmann, T. Tušar, D. Brockhoff. COCO: A Platform for Comparing Continuous Optimizers in a Black-Box Setting, Optimization Methods and Software, 36(1), pp. 114-144, 2021”

3)	The performance of the algorithm heavily depends on the efficiency of the latent slice sampler  applied to the nascent minima distribution $m_k$.

**Questions:**

1)	How to calculate the denominator of Equations (2) and (3) for a general black-box $f(x)$?

2)	In Algorithm 2, why we don’t use only a single large value for $k$? In Line 3, is $x^{(0)}$ that same for every t-th iteration? I don’t see any interaction between iterations; that is, the information of $x^{(1)}, …, x^{(N)}$  is not reused in next steps.

3)	In Line 7 of Algorithm 2, $x$ is a vector, $x_j$ is a scalar, what does it mean by $x < x_j$?

4)	What is the impact of selecting $k$ in Eq. (3) on the solution quality? What should the appropriate value for $k$?

5)	In Algorithm 2, why we need to fix $T = 200$?

6)	What are stopping criteria for ProGO and other baseline methods used in Section 4?

---

> ### Author Response · Authors · 2023-11-21
> **Response to Reviewer dC7R - Weaknesses**
>
> > W1: “In order to apply the proposed algorithm, the optimal function value
>  is supposed to be known. The assumption is not true for many practical applications.”
>
> We do not need the knowledge of optimal function value $f^*$, and we have demonstrated this in Section 3 (also see the Remark 1 that we included right after defining the nascent minima distribution). We used $f^*$ only for the ease of theoretical proofs, but as we pointed out in our remark, the $exp(-f^*)$ cancels out in the numerator and denomination (and so in practice we do not need to know the $f*$. The requirement for all inputs is listed in “Input” of Algorithm 2. We only use $f^*$ when evaluating experimental performance on synthetic functions in Section 4. We will make it clearer for the modified version.
>
> > W2: “The algorithm is only tested on two test instances, which can be efficiently solved by a number of existing derivative-free optimization algorithms.”
>
> We acknowledge the importance of conducting extensive experiments to validate its performance advantages as recommended. We have incorporated additional examples in Appendix B and intend to explore further applications in future research.
>
> Additionally, our initial algorithm testing, while limited to two instances, is notable for its high-dimensional setting, spanning 1000 dimensions. This departure from the typical 100 dimensions in other studies is worth emphasizing.
>
>
> > W3: “The performance of the algorithm heavily depends on the efficiency of the latent slice sampler applied to the nascent minima distribution”
>
> We appreciate the reviewer's insights into the significant role of the latent slice sampler (LSS). LSS is a part of the slice samplers (SS) family, which is well-known for its robust convergence efficiency: [1] demonstrated their nearly geometric ergodic behavior, and [2] outlined conditions for SS to achieve uniform ergodicity. Extending the SS framework, [3] describes the detailed rationale and practical convergence efficiency of LSS.
>
> [1] Roberts, Gareth O., and Jeffrey S. Rosenthal. "Convergence of slice sampler Markov chains." Journal of the Royal Statistical Society Series B: Statistical Methodology 61.3 (1999): 643-660.
> [2] Mira, Antonietta, and Luke Tierney. "Efficiency and convergence properties of slice samplers." Scandinavian Journal of Statistics 29.1 (2002): 1-12.
> [3] Li, Yanxin, and Stephen G. Walker. "A latent slice sampling algorithm." Computational Statistics & Data Analysis 179 (2023): 107652.

---

> ### Author Response · Authors · 2023-11-21
> **Response to Reviewer dC7R - Questions**
>
> > Q1: “How to calculate the denominator of Equations (2) and (3) for a general black-box target function”
>
> We want to emphasize that there's no need to compute the denominator in the equation (2). This is because $m_k(x) \propto {e^{-kf(x)} \cdot \pi(x) }$, and the integral serves as a constant. As it’s well-known within the MCMC literature, the normalizing constant of the density is not required to generate samples from the path of the Markov chain (by using the LSS). To clarify, for any black-box target function, all that are required are MCMC samples generated from the kennel of the density  ${e^{-kf(x)} \cdot \pi(x) }$ to approximate the minima distribution in equation (2). Hence, there’s no need to compute the denominator to implement the LSS.
>
>
> > Q2: “In Algorithm 2, why we don’t use only a single large value for k ? In Line 3, is that same for every t-th iteration? I don’t see any interaction between iterations; that is, the information of is not reused in next steps.”
>
> In Algorithm 2, employing a single large k within the exponential can result in numerical instability in the sampling algorithm. Our goal is to find k sufficiently large enough that makes LSS numerically stable until the convergence of the minima is numerically achieved. On the other hand, the latent slice sampler demonstrates a geometric rate of convergence even when with smaller values of $k$. Despite no direct interaction between iterations, the algorithm can achieve the desired results effectively. This justifies our approach of conducting a sequential search for the optimal k, considering both practical and theoretical considerations.
>
> > Q3: “In Line 7 of Algorithm 2, x is a vector, xj is a scalar, what does it mean by x < xj?”
>
> Thank you for bringing this to our attention. We have already corrected it to $x_j$ < $x_j^{(t-1)}$.
>
> > Q4: “What is the impact of selecting k in Eq. (3) on the solution quality? What should the appropriate value for k?”
>
> To illustrate the effect of k, we provided a one-dimensional illustration in Figure 2, which visualized the impact of increasing k from 0 to 1, 3, and 9. As k increases, the density of $m_k(x)$ converges towards the global optima of $f(x)$, resulting in more accurate sampling around the global optima.
>
> Determining the appropriate value of k depends on the specific characteristics of the target function, including its value and dimension. Hence, throughout our sequential search, we explored a wide range of k values, covering from $k=5$ to $k=5e^{200}$. As our method is shown to converge theoretically to the minima, our goal here is to approximate the minima by slowly increasing the k (maintaining the stability of the LSS) until a numerical convergence is observed.
>
>
>
>
> > Q5: “In Algorithm 2, why we need to fix T=200?”
>
> We chose the value 200 because we found in preliminary results that our method's regrets had already reached an extremely small value of $1\times10^{-17}$, far less than the threshold that can be accurately represented by floating-point numbers. Admittedly, the chosen value is a bit ad hoc, but this choice is purely driven by our experiments.
>
>
> > Q6: “What are stopping criteria for ProGO and other baseline methods used in Section 4?”
>
> The stopping criteria is 200 iterations for each of the methods, aligned with the settings in [1].
>
> [1] Shu, Yao, et al. "Zeroth-Order Optimization with Trajectory-Informed Derivative Estimation." The Eleventh International Conference on Learning Representations. 2022.

---

### Official Review · Reviewer_vc8T · 2023-10-31

**Soundness:** 3 good
**Presentation:** 3 good
**Contribution:** 2 fair
**Rating:** 5
**Confidence:** 3

**Summary:**

The paper proposes a gradient-free numerical algorithm to solve the global optimization problem. The algorithm is probabilistic, designed to sample from a sequence of probability distributions that converge to a distribution supported on the global minima set. The sequence of distributions $m_k(x)$ is constructed by weighing a uniform probability distribution by exponential of the objective function $\exp(-kf(x)$ for increasing $k$. The proposed algorithm is illustrated and compared on several numerical examples, demonstrating its efficiency and accuracy.

**Strengths:**

- The paper is nicely written with clear presentation and explanations of the contributions.
- The proposed algorithm is supported by theoretical asymptotic convergence results.
- The numerical experiments report strong support for the efficiency and accuracy of the algorithm in comparison with several approaches.

**Weaknesses:**

- Although the algorithm is based on the convergence of m_k, the rationale behind the proposed sampling procedure for m_k is not clear.
 -There is no non-asymptotic analysis that relates the number of function evaluations with the optimality gap.
- Although theoretical result is nice, it does not explain why this algorithm performs better than alternative approaches. As a result, two numerical experiments might not be sufficient to support the paper's claim.
- The paper misses discussing model-based optimization algorithms as in [1]. Similar theoretical convergence results exists in this and other papers. For example, see Thm. 3 in [2]. The convergence holds under weaker assumptions for the objective function: lower-semicontinuous instead of continuous; and the strong separable condition seems to hold (see the discussion in Appendix C of [2]).

[1] Hu, Jiaqiao, et al. "A survey of some model-based methods for global optimization." Optimization, Control, and Applications of Stochastic Systems (2012): 157-179.

[2] Zhang, Chi, Amirhossein Taghvaei, and Prashant G. Mehta. "A mean-field optimal control formulation for global optimization." IEEE Transactions on Automatic Control 64.1 (2018): 282-289.

**Questions:**

Please see my comments above.

---

> ### Author Response · Authors · 2023-11-21
> **Response to Reviewer vc8T**
>
> >W1-1: “The rationale behind the proposed sampling procedure for $m_k$ is not clear.”
>
> The rationale for our sampling procedure is based on the fact that the mode (and also the mean) of the density $m_k(x)$ converges to the minima when it is unique. Consequently, by sampling random variates from this density, we estimate the mode and the mean using Monte Carlo methods, as exact analytical calculations are often impractical. A better understanding of this rationale might be gained by referring to our one dimensional Illustration example presented in Figure 2 within Section 3.1.
>
> In addition, following up with your question, we have made an attempt to make our rationale clearer at the beginning of Section 3. Thank you for your feedback.
>
>
>
> >W1-2: “ There is no non-asymptotic analysis that relates the number of function evaluations with the optimality gap”
>
> The family of slice samplers (SS) is well-known for their geometric rates of convergence properties. For instance, [1] demonstrated the near-geometric ergodic behavior of SS, and [2] provided conditions for SS to achieve uniform ergodicity. As an extension of SS, the detailed algorithm of latent slice sampler (LSS) and its practical convergence have been discussed in [3]. Our method of approximating the minima is based on the stochastic sampling from the minima distribution $m_k(x)$ using LSS. Hence, obtaining non-asymptotic bounds is a non-trivial task, which we will explore in future research.
>
> [1] Roberts, Gareth O., and Jeffrey S. Rosenthal. "Convergence of slice sampler Markov chains." Journal of the Royal Statistical Society Series B: Statistical Methodology 61.3 (1999): 643-660.
>
> [2] Mira, Antonietta, and Luke Tierney. "Efficiency and convergence properties of slice samplers." Scandinavian Journal of Statistics 29.1 (2002): 1-12.
>
> [3] Li, Yanxin, and Stephen G. Walker. "A latent slice sampling algorithm." Computational Statistics & Data Analysis 179 (2023): 107652.
>
> > W2: “It does not explain why this algorithm performs better than alternative approaches”
>
> There are very few algorithms that are comparable to our proposed method when the target function is assumed to be a continuous function (e.g., our method doesn’t make use of gradient vectors). Besides, our method provides a theoretical convergence guarantee for non-convex functions with a global optima.  It's important to note that our approach requires only the continuity of the target function f(x), and the results are proved essentially under very mild conditions, avoiding the need for any gradient or smoothness conditions (e.g., Lipstiz, etc.), which are typically prerequisites in most well-known methods. Also, most existing methods are based on hot (random) starting values, but our method doesn't require that, which is one of the novelties of our proposed method compared to other existing optimization methods. In essence, most other existing methods are not strictly comparable for the above reasons.
>
>
> > W3: “The paper misses discussing model-based optimization algorithms as in [1]. Similar theoretical convergence results exists in this and other papers. For example, see Thm. 3 in [2]. The convergence holds under weaker assumptions for the objective function: lower-semicontinuous instead of continuous; and the strong separable condition seems to hold (see the discussion in Appendix C of [2]).”
>
> Our approach stands out significantly different from model-based methodologies. Model-based methods typically provide approximations of optimality through a two-step process: starting with an initial modeling assumption that approximates the target function and subsequently deriving estimators based on the assumed model. Our proposed method requires no such sugrrorgate function to approximate the target function but rather is based on the given target function.
>
> In reference to [2], it explores the Bayes convergence of a model-based solutions, particularly in the context of ODE-PDE problems, which exhibit closed-form solutions akin to the minima distribution. However, the proof of Theorem 3 [2] in Appendix C relies on the assumption of second continuous differentiability on a compact set. In contrast, our method operates under a more relaxed continuity assumption on a non-compact set, without the need for gradient information, showcasing a novel approach in comparison.
>
> [1] Hu, Jiaqiao, et al. "A survey of some model-based methods for global optimization." Optimization, Control, and Applications of Stochastic Systems (2012): 157-179.
>
> [2] Zhang, Chi, Amirhossein Taghvaei, and Prashant G. Mehta. "A mean-field optimal control formulation for global optimization." IEEE Transactions on Automatic Control 64.1 (2018): 282-289.

---

### Meta-Review · Area_Chair_QB6W · 2023-12-07

**Metareview:**

The paper proposes a probabilistic method for gradient-free global optimization for continuous functions.

The paper is well written and sound, but the reviewers have raised severe doubts about the novelty and experimental evaluation.

**Justification For Why Not Higher Score:**

The reviewers have raised severe doubts about the novelty and experimental evaluation.

**Justification For Why Not Lower Score:**

NA

---

### Decision · Program_Chairs · 2024-01-16

Reject